# Training-Free Vector Quantization via Gaussian VAEs

**Tongda Xu** [* 1 2 3]   **Wendi Zheng** [2 3]   **Jiajun He** [4]   **José Miguel Hernández-Lobato** [4]   **Yan Wang** [1]   **Ya-Qin Zhang** [1]   **Jie Tang** [3]

## Abstract

Vector-quantized variational autoencoders (VQ-VAEs) are discrete autoencoders that compress images into discrete tokens. However, they are difficult to train due to discretization. In this paper, we propose a simple yet effective technique dubbed **Gaussian Quant (GQ)**, which first trains a Gaussian VAE under certain constraints and then converts it into a VQ-VAE without additional training. For conversion, GQ generates random Gaussian noise as a codebook and finds the closest noise vector to the posterior mean. Theoretically, we prove that when the logarithm of the codebook size exceeds the bits-back coding rate of the Gaussian VAE, a small quantization error is guaranteed. Practically, we propose a heuristic to train Gaussian VAEs for effective conversion, named the target divergence constraint (TDC). Empirically, we show that GQ outperforms previous VQ-VAEs, such as VQGAN, FSQ, LFQ, and BSQ, on both UNet and ViT architectures. Furthermore, TDC also improves previous Gaussian VAE discretization methods, such as TokenBridge. The source code is provided in https://github.com/tongdaxu/VQ-V AE-from-Gaussian-VAE.

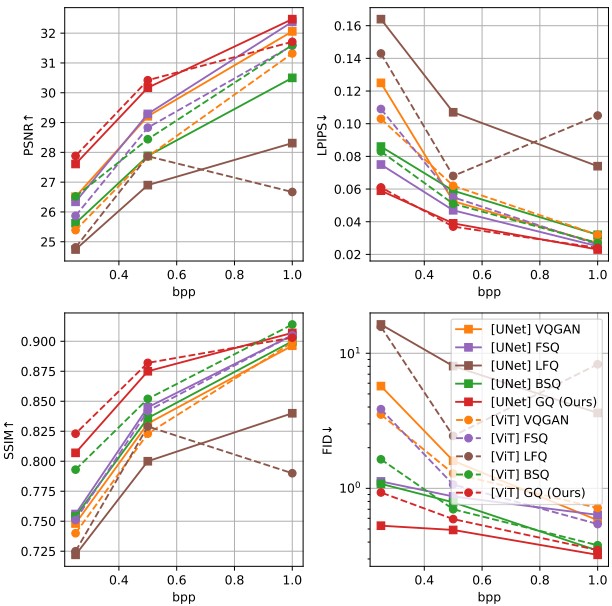

*Figure 1.* The rate-distortion performance on the ImageNet dataset demonstrates that GQ outperforms previous VQ-VAEs on both UNet and ViT architectures.

## 1. Introduction

Vector-quantized variational autoencoders (VQ-VAEs) (Van Den Oord et al., 2017) compress images into discrete tokens and are fundamental to autoregressive generative models (Esser et al., 2021; Chang et al., 2022; Yu et al., 2023; Sun et al., 2024b). However, VQ-VAEs are difficult to train: the

encoding process is non-differentiable, and issues such as codebook collapse often arise (Sønderby et al., 2017). As a result, special techniques are required, including commitment loss (Van Den Oord et al., 2017), expectation maximization (EM) (Roy et al., 2018), Gumbel-Softmax (Jang et al., 2016; Maddison et al., 2016; Sønderby et al., 2017), and entropy loss (Yu et al., 2023; Zhao et al., 2024).

In this paper, we circumvent the difficulty of training VQ-VAEs by not training them at all. Specifically, we propose **Gaussian Quant (GQ)**, which first trains a constrained Gaussian VAE and then converts it into a VQ-VAE without additional training. For conversion, GQ generates a codebook of one-dimensional Gaussian noise and, for each posterior dimension, select the entry closest to the posterior mean. We show theoretically that when the logarithm of the codebook size exceeds the bits-back coding rate (Hinton & Van Camp, 1993; Townsend et al., 2019) of the Gaussian VAE, the quantization error is small, providing a principled guideline for codebook size selection.

---

*The work was done when Tongda Xu was a full-time intern with Zhipu AI. [1]AIR, Tsinghua University [2]Zhipu AI [3]CST, Tsinghua University [4]University of Cambridge. Correspondence to: Yan Wang <wangyan@air.tsinghua.edu.cn>, Jie Tang <jietang@tsinghua.edu.cn>.

*Proceedings of the 43rd International Conference on Machine Learning*, Seoul, South Korea. PMLR 306, 2026. Copyright 2026 by the author(s).

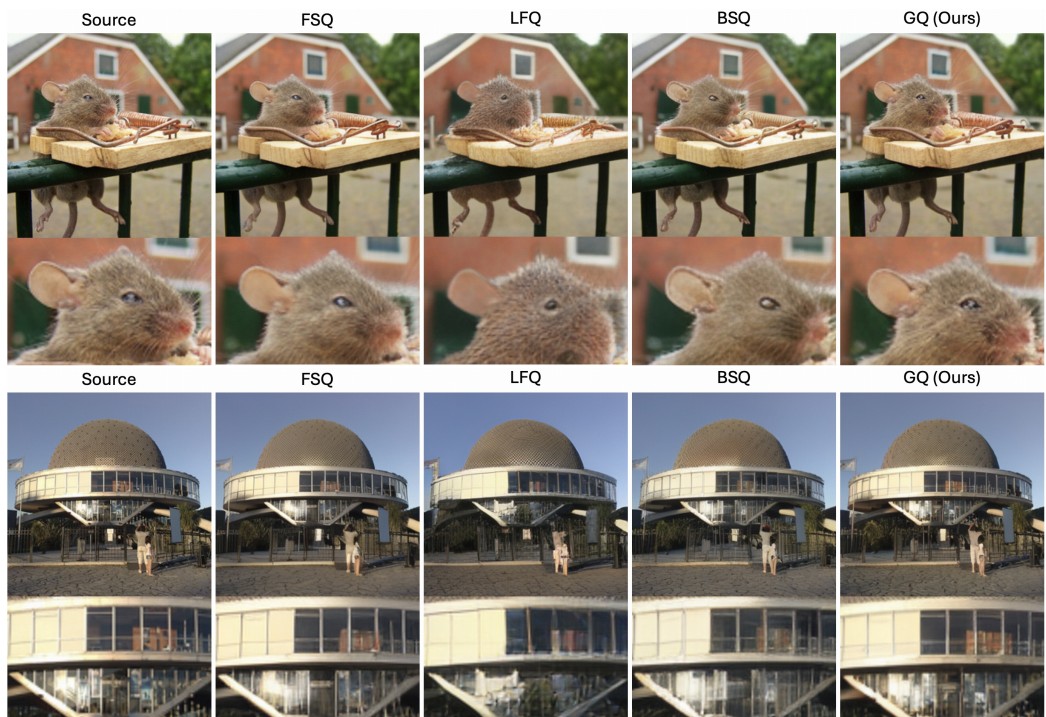

*Figure 2.* Qualitative results on ImageNet dataset at 0.25 bits-per-pixel (bpp). GQ has the most visually pleasing reconstruction.

To train a Gaussian VAE for effective conversion, we introduce the **target divergence constraint (TDC)**, which encourages the Gaussian VAE to achieve the same Kullback–Leibler (KL) divergence for each dimension, corresponding to the bits-back coding bitrate. Empirically, we demonstrate that GQ with Gaussian VAE trained by TDC, outperforms previous VQ-VAEs such as VQGAN, FSQ, LFQ, and BSQ (Van Den Oord et al., 2017; Mentzer et al., 2023; Yu et al., 2023; Zhao et al., 2024) in terms of reconstruction, on UNet and ViT backbones. Additionally, we show that TDC improves previous Gaussian VAE discretization methods, such as TokenBridge (Wang et al., 2025).

Our contributions are summarized as follows:

- (Section 3.1) We propose Gaussian Quant (GQ), a simple yet effective approach that constructs a VQ-VAE by training a constrained Gaussian VAE and converting it into a VQ-VAE without additional training.

- (Section 3.2) We theoretically show that when the GQ codebook size matches the bits-back coding rate of the Gaussian VAE, the conversion error remains small.

- (Section 3.3) We introduce the target divergence constraint (TDC) to train Gaussian VAEs for effective conversion, and show that GQ outperforms prior VQ-VAEs on both UNet and ViT architectures.

- (Section 3.6) We show that TDC improves TokenBridge, an existing Gaussian VAE conversion methods.

## 2. Preliminaries

We denote the original image as $X$, the latent as $Z$, the encoder as $f(\cdot)$, and the decoder as $g(\cdot)$. We use $\log(\cdot)$ for the natural logarithm and $D_{KL}(\cdot||\cdot)$ for KL divergence. We denote the bits-back coding bitrate as $R_i = D_{KL}(q(Z_i|X)||\mathcal{N}(0, I))$. We denote the base-2 logarithm as $\log_2(\cdot)$, with the corresponding bitrate $R_i^{\text{bits}} = R_i/\log 2$.

### 2.1. Vector Quantized Variational Autoencoder

VQ-VAE (Van Den Oord et al., 2017) encodes images into integer tokens. For autoregressive generation, it uses deterministic encoding and a shared codebook. Specifically, VQ-VAE maintains a codebook $c_{1:K}$ of size $K$ with bitrate $\log K$. For each dimension $i$, the encoder output $f(x)_i$ is quantized to the closest codeword $c_j$ in $c_{1:K}$. Denoting distortion as $\Delta(\cdot, \cdot)$, the VQ-VAE objective is the rate-distortion function weighted by a Lagrange multiplier $\lambda$:

$$\mathcal{L}_{VQ} = \lambda \underbrace{\log K}_{\text{bitrate}} + \underbrace{\mathbb{E}[\Delta(X, g(\hat{z}))]}_{\text{distortion}} + \mathcal{L}_{Reg},$$
$$\hat{z}_i = \arg\min_{c_j \in c_{1:K}} ||f(x)_i - c_j||, \qquad (1)$$

where $c_{1:K}$ is the learned codebook, and $\mathcal{L}_{\text{Reg}}$ is a regularization term that ensures VQ-VAE convergence, such as commitment loss, codebook loss (Van Den Oord et al., 2017), and Gumbel-Softmax loss (Sønderby et al., 2017).

## 2.2. Gaussian VAE and Bits-Back Coding

The Gaussian VAE is a special VAE (Kingma et al., 2013) with a prior $\mathcal{N}(0, I)$ and a fully factorized Gaussian posterior $q(Z|X)$. Its encoding simply involves sampling each latent dimension $z_i \sim q(Z_i|X)$. Assuming $\log p(X|Z = z) \propto (1/\lambda)\Delta(X, g(z))$, the negative evidence lower bound (ELBO) of the Gaussian VAE corresponds to a rate-distortion function, consisting of a bits-back coding bitrate term and a distortion term:

$$\mathcal{L}_{VAE} = \lambda \underbrace{D_{KL}(q(Z|X)\|\mathcal{N}(0,1))}_{\text{bits-back coding bitrate}} + \underbrace{\mathbb{E}[\Delta(X, g(z))]}_{\text{distortion}},$$

$$z_i \sim q(Z_i|X = x) = \mathcal{N}(\mu_i, \sigma_i^2), i = 1 \ldots d. \quad (2)$$

The **bits-back coding bitrate** of $Z_i$ is defined as $R_i = D_{KL}(q(Z_i|X)\|\mathcal{N}(0, I))$ (Hinton & Van Camp, 1993), because when compressing $X$ losslessly, $z_i$ can be communicated using $R_i$ nats to arbitrary precision.

# 3. Gaussian Quant: Vector Quantization using Gaussian VAE

We propose an alternative approach to obtain a VQ-VAE: first train a constrained Gaussian VAE, then convert it into a VQ-VAE.

## 3.1. Converting a Gaussian VAE into a VQ-VAE

Given a Gaussian VAE, we convert it into a VQ-VAE by generating one-dimensional Gaussian noise as a codebook and quantizing the posterior mean independently for each latent dimension. The codebook is fixed once generated. Because it consists entirely of Gaussian samples, we call this approach **Gaussian Quant (GQ)**. Specifically, we randomly generate a codebook $c_{1:K} \sim \mathcal{N}(0, 1)$ shared across all dimensions. Then, for each dimension $i$, we select the codeword $c_j$ closest to the posterior mean $\mu_i$ and denote the quantized value as $\hat{z}_i$:

$$\hat{z}_i = \arg \min_{c_j \in c_{1:K}} \|\mu_i - c_j\|, \text{ where } c_{1:K} \sim \mathcal{N}(0, 1). \quad (3)$$

## 3.2. Theoretical Relationship between the Codebook Size and Quantization Error

Why GQ works and how to choose $K$ are not obvious. Theoretically, we show that GQ preserves the rate-distortion properties of the Gaussian VAE: when the codebook bitrate $\log K$ matches the bits-back coding rate $R_i$ of the Gaussian VAE, the quantization error is small. More specifically, we denote the event $|\hat{z}_i - \mu_i| \geq \sigma_i$ as large quantization error. Then, we show that the probability of a large quantization error decays doubly exponentially as the codebook bitrate $\log K$ exceeds the bits-back coding rate $R_i$.

**Theorem 3.1.** *Denote the mean and standard deviation of $q(Z_i|X = x)$ as $\mu_i$ and $\sigma_i$, respectively. We assume $|\mu_i\sigma_i| \leq c_1$ and $|\mu_i| + |\sigma_i| \leq c_2$. Given a fixed $R_i = D_{KL}(q(Z_i|X)\|\mathcal{N}(0,1))$, the probability of a quantization error $|\hat{z}_i - \mu_i| \geq \sigma_i$ decays doubly exponentially with the number of nats $t$ by which the codebook bitrate $\log K$ exceeds the bits-back coding rate. That is,*

$$\text{when } \log K = R_i + t,$$

$$\Pr\{|\hat{z}_i - \mu_i| \geq \sigma_i\} \leq \exp\left(-e^t \sqrt{\frac{2}{\pi}} e^{-c_1 - 0.5}\right). \quad (4)$$

Conversely, when the codebook bitrate $\log K$ is smaller than the bits-back coding rate $R_i$, the probability of a large quantization error increases exponentially toward 1.

**Theorem 3.2.** *Following Theorem 3.1, the probability of a quantization error $|\hat{z}_i - \mu_i| \geq \sigma_i$ increases exponentially with the number of nats $t$ by which the codebook bitrate $\log K$ falls below the bits-back coding rate. That is,*

$$\text{when } \log K = R_i - t,$$

$$\Pr\{|\hat{z}_i - \mu_i| \geq \sigma_i\} \geq 1 - e^{-t}\sqrt{\frac{2}{\pi}} e^{0.5c_2^2 - 0.5}. \quad (5)$$

Theorems 3.1 and 3.2 provide a principled guideline for choosing $K$: $\log K$ should be close to the bits-back rate $R_i$. In practice, setting $\log_2 K = \lceil R_i^{\text{bits}} \rceil$ typically yields sufficiently small reconstruction error, where $\lceil \cdot \rceil$ denotes rounding. Using a larger $K$ offers no additional benefit, while a smaller $K$ significantly increases the error.

## 3.3. Training a Conversion-ready Gaussian VAE with Target Divergence Constraint

Now that we know how to convert a Gaussian VAE into a VQ-VAE, we discuss how to train a Gaussian VAE for effective conversion. To construct a VQ-VAE with a specific codebook size $K$, we need to train the Gaussian VAE so that $R_i$ is close to $\log K$ for all dimensions $i = 1, \ldots, d$.

Previous works such as MIRACLE and HiFiC (Flamich et al., 2020; Mentzer et al., 2020) have developed heuristic that limits the mean of $R_i$ across all dimensions. More specifically, they augment the original target of VAE in Eq. 2 by updating $\lambda$ according to the relationship between target bitrate $R^*$ and the mean of actual bitrate $R_i$ with update rate $\beta$:

$$\mathcal{L}_{HiFiC} = \lambda \sum_{i=1}^{d} R_i + \mathbb{E}[\Delta(X, g(z))],$$

$$\lambda = \beta\lambda \text{ if } \text{mean}_i\{R_i\} > R^* \text{ else } \beta^{-1}\lambda.$$

$$(6)$$

To train a Gaussian VAE under stricter per-dimension constraint, we propose a heuristic **Target Divergence Constraint (TDC)**, which is an extension of previous $\mathcal{L}_{HiFiC}$ heuristic with per-dimension rate control. Specifically, we set the target $R_i$ to $\log K$. For each dimension $i$, we apply a larger penalty if $R_i$ exceeds $\log K + \alpha$ and a smaller penalty if $R_i$ falls below $\log K - \alpha$, using different $\lambda$ values for each case. Here, $\alpha$ is a hyperparameter controlling the threshold:

$$\mathcal{L}_{TDC} = \sum_{i=1}^{d} \lambda_i R_i + \mathbb{E}[\Delta(X, g(z))],$$

$$\text{where } \lambda_i = \begin{cases} \lambda_{\min}, & R_i < \log K - \alpha, \\ \lambda_{\text{mean}}, & R_i \in [\log K \pm \alpha], \\ \lambda_{\max}, & R_i > \log K + \alpha. \end{cases} \quad (7)$$

To determine $\lambda_{\min}, \lambda_{\text{mean}}, \lambda_{\max}$, we initialize them to 1 and update them according to the relationship between the statistics of $R_i$ and $\log K \pm \alpha$ during each gradient descent step:

$$\lambda_{\min} = \beta \lambda_{\min} \text{ if } \min_i\{R_i\} > \log K - \alpha \text{ else } \beta^{-1}\lambda_{\min},$$

$$\lambda_{\text{mean}} = \beta \lambda_{\text{mean}} \text{ if } \text{mean}_i\{R_i\} > \log K \text{ else } \beta^{-1}\lambda_{\text{mean}},$$

$$\lambda_{\max} = \beta \lambda_{\max} \text{ if } \max_i\{R_i\} > \log K + \alpha \text{ else } \beta^{-1}\lambda_{\max},$$

where $\beta$ is a hyperparameter controlling the update speed. To avoid numerical issues, we clip $\lambda_{\min}, \lambda_{\text{mean}}, \lambda_{\max}$ to the range $[10^{-3}, 10^3]$ after each update.

### 3.4. Supporting Multi-dimensional Codebook

The vanilla VQ-VAE has three key parameters: codebook size $K$, codebook dimension $m$, and number of tokens $N$. Now GQ is able to construct VQ-VAE with codebook dimension $m = 1$, we can extend GQ to codebook dimension $m > 1$. Specifically, given a Gaussian VAE, we can quantize $m$ latents into a single large token with codebook size $\log_2 K = \left\lceil \sum_{k=i}^{i+m} R_k^{\text{bits}} \right\rceil$. During quantization, we minimize the $\sigma_i$-weighted distance to find nearest neighbour, with the codebook $c_{1:K}$ sampled from $m$ dimensional factorized standard Gaussian:

$$\hat{z}_{i:i+m} = \arg \min_{c_j \in c_{1:K}} ||(\mu_{i:i+m} - c_j)/\sigma_{i:i+m}||,$$

$$\text{where } c_{1:K} \sim \mathcal{N}(0, I_m). \quad (8)$$

However, for low-bitrate cases, this vanilla $m$-dimensional conversion leads to codebook collapse. This occurs because $|\mu_i|$ is bounded by $\sqrt{2R_i}$ (see Lemma A.1), so when $R_i$ is small, codebook vectors far from 0 are never selected. To address this, we introduce a regularization term weighted by $\omega$ that encourages the selection of $c_j$ farther from 0:

$$\hat{z}_{i:i+m} = \arg \min_{c_j \in c_{1:K}} ||(\mu_{i:i+m} - c_j)/\sigma_{i:i+m}|| - \omega||c_j||.$$

Now we have the conversion method for multi-dimensional codebook, we consider how to train a Gaussian VAE for multi-dimensional codebook. Specifically, we modify the TDC training target to account for the relationship between the sum of $m$ bitrates $\sum_{k=i}^{i+m} R_k$ and $\log_2 K \pm \alpha$. The detailed TDC update rule is:

$$\mathcal{L}_{TDC}^m = \sum_{j=1}^{d//m} \lambda_j \sum_{k=jm}^{jm+m} R_k + \mathbb{E}[\Delta(X, g(z))],$$

$$\text{where } \lambda_i = \begin{cases} \lambda_{\min}, & \sum_{k=jm}^{jm+m} R_k < \log_2 K - \alpha, \\ \lambda_{\text{mean}}, & \sum_{k=jm}^{jm+m} R_k \in [\log_2 K \pm \alpha], \\ \lambda_{\max}, & \sum_{k=jm}^{jm+m} R_k > \log_2 K + \alpha. \end{cases}$$
$$(9)$$

And the update rule for $\lambda$s can be modified in a similar way:

$$\lambda_{\min} = \beta \lambda_{\min} \text{ if } \min_j\{\sum_{k=jm}^{jm+m} R_k\} > \log_2 K - \alpha \text{ else } \beta^{-1}\lambda_{\min},$$

$$\lambda_{\text{mean}} = \beta \lambda_{\text{mean}} \text{ if } \text{mean}_j\{\sum_{k=jm}^{jm+m} R_k\} > \log_2 K \text{ else } \beta^{-1}\lambda_{\text{mean}},$$

$$\lambda_{\max} = \beta \lambda_{\max} \text{ if } \max_j\{\sum_{k=jm}^{jm+m} R_k\} > \log_2 K + \alpha \text{ else } \beta^{-1}\lambda_{\max}.$$

### 3.5. Practical Guidelines

To summarize, GQ has the same interface as a VQ-VAE and constructs a VQ-VAE with codebook size $K$, codebook dimension $m$, and number of tokens $N$ with two steps:

- First, GQ trains a Gaussian VAE with latent dimension $d = mN$, using the TDC constraint with target $\log 3K$ as in Eq. 9.

- Next, GQ converts this constrained Gaussian VAE into a VQ-VAE using Eq. 8. The codebook $c_{1:K}$ is fixed once generated.

### 3.6. Improving TokenBridge with Target Divergence Constraint

TokenBridge (Wang et al., 2025) also converts a pre-trained Gaussian VAE into a VQ-VAE. It adopts the Post-Training Quantization (PTQ) technique from model compression, treating the latent as model parameters for discretization. TokenBridge uses a fixed codebook of $2^K$ centroids from a Gaussian distribution and quantizes the posterior by selecting the closest centroid. However, it directly quantizes a vanilla Gaussian VAE without constraining the KL divergence of each dimension, leading to suboptimal rate-distortion performance.

We can improve TokenBridge using TDC. Its quantization centers correspond to equal-probability partition centers of $\mathcal{N}(0, 1)$, which is a special case of GQ with an evenly distributed codebook $c_{1:K}$. Therefore, the number of PTQ bits should also match $R_i$, and TDC can enhance TokenBridge performance (see Table 3).

*Table 1.* Quantitative results on the ImageNet dataset, comparing different VQ methods using the same model architecture. Our GQ outperforms other VQ methods on both UNet and ViT architectures. **Bold**: best. Gray: continuous Gaussian VAE.

| Method | bpp ($K \times N$) | UNet based | | | | ViT based | | | |
|---|---|---|---|---|---|---|---|---|---|
| | | PSNR↑ | LPIPS↓ | SSIM↑ | rFID↓ | PSNR↑ | LPIPS↓ | SSIM↑ | rFID↓ |
| Gaussian VAE | ≈ 0.25 (-) | 28.60 | 0.047 | 0.849 | 0.556 | 28.95 | 0.045 | 0.851 | 0.672 |
| Gaussian VAE (w/ TDC) | | 28.05 | 0.053 | 0.829 | 0.535 | 28.44 | 0.054 | 0.837 | 0.793 |
| VQGAN (Esser et al., 2021) | | 26.51 | 0.125 | 0.748 | 5.714 | 25.39 | 0.103 | 0.740 | 3.518 |
| FSQ (Mentzer et al., 2023) | | 26.34 | 0.075 | 0.756 | 1.125 | 25.87 | 0.109 | 0.751 | 3.856 |
| LFQ (Yu et al., 2023) | 0.25 ($2^{16} \times 1024$) | 24.74 | 0.164 | 0.722 | 16.337 | 24.81 | 0.143 | 0.725 | 15.71 |
| BSQ (Zhao et al., 2024) | | 25.62 | 0.086 | 0.754 | 1.080 | 26.52 | 0.083 | 0.793 | 1.649 |
| GQ (Ours) | | **27.61** | **0.059** | **0.807** | **0.529** | **27.88** | **0.061** | **0.823** | **0.932** |
| Gaussian VAE | ≈ 0.50 (-) | 31.18 | 0.028 | 0.889 | 0.389 | 31.64 | 0.030 | 0.900 | 0.867 |
| Gaussian VAE (w/ TDC) | | 30.84 | 0.035 | 0.884 | 0.549 | 31.18 | 0.033 | 0.896 | 0.757 |
| VQGAN (Esser et al., 2021) | | 29.21 | 0.052 | 0.831 | 1.600 | 27.86 | 0.062 | 0.823 | 1.228 |
| FSQ (Mentzer et al., 2023) | | 29.29 | 0.047 | 0.845 | 0.871 | 28.83 | 0.055 | 0.842 | 1.067 |
| LFQ (Yu et al., 2023) | 0.50 ($2^{16} \times 2048$) | 26.90 | 0.107 | 0.800 | 8.035 | 27.87 | 0.068 | 0.829 | 2.444 |
| BSQ (Zhao et al., 2024) | | 27.88 | 0.059 | 0.836 | 0.788 | 28.44 | 0.051 | 0.852 | 0.700 |
| GQ (Ours) | | **30.17** | **0.039** | **0.875** | **0.492** | **30.42** | **0.037** | **0.882** | **0.592** |
| Gaussian VAE | ≈ 1.00 (-) | 32.73 | 0.022 | 0.910 | 0.490 | 32.28 | 0.023 | 0.910 | 0.469 |
| Gaussian VAE (w/ TDC) | | 32.75 | 0.021 | 0.910 | 0.294 | 32.05 | 0.023 | 0.909 | 0.397 |
| VQGAN (Esser et al., 2021) | | 32.06 | 0.026 | 0.896 | 0.580 | 31.32 | 0.032 | 0.899 | 0.716 |
| FSQ (Mentzer et al., 2023) | | 32.38 | 0.025 | 0.905 | 0.636 | 31.58 | 0.026 | 0.905 | 0.544 |
| LFQ (Yu et al., 2023) | 1.00 ($2^{16} \times 4096$) | 28.31 | 0.074 | 0.840 | 3.617 | 26.67 | 0.105 | 0.790 | 8.288 |
| BSQ (Zhao et al., 2024) | | 30.50 | 0.032 | 0.900 | 0.346 | 31.60 | 0.027 | **0.914** | 0.379 |
| GQ (Ours) | | **32.47** | **0.023** | **0.907** | **0.322** | **31.71** | **0.024** | 0.903 | **0.349** |

# 4. Experimental Results

## 4.1. Experimental Setup

**Models and Baselines** For image reconstruction, we use two representative autoencoder architectures: **UNet** from Stable Diffusion 3 (Esser et al., 2024), and **ViT** from BSQ (Zhao et al., 2024). For VQ baselines, we train vanilla **VQGAN** (Esser et al., 2021) and several variants, including **FSQ** (Mentzer et al., 2023), **LFQ** (Yu et al., 2023), and **BSQ** (Zhao et al., 2024) on the same model architecture. Additionally, we compare to pre-trained VQ-VAEs such as **VQGAN-Taming** (Esser et al., 2021), **VQGAN-SD** (Rombach et al., 2022), **Llama-Gen** (Sun et al., 2024b), **FlowMo** (Sargent et al., 2025), **BSQ** (Zhao et al., 2024), **OpenMagViT-V2** (Luo et al., 2024), and conversion methods like **TokenBridge** and **ReVQ**. We further show that our TDC technique improves TokenBridge (Wang et al., 2025), a previous training-free approach for converting Gaussian VAEs into VQ-VAEs. For image generation, we employ the Llama transformer (Touvron et al., 2023; Shi et al., 2024a).

**Datasets, Bitrates, and Metrics** We use the **ImageNet** (Deng et al., 2009) training split for training, and the **ImageNet** and **COCO** (Lin et al., 2014) validation splits for testing. For reconstruction and generation experiments, images are resized to $256 \times 256$ and $128 \times 128$, respectively. For bitrates, we evaluate reconstruction performance using **codebook sizes K** from $2^{14}$ to $2^{18}$ and **number of token N**

of 256, 1024, 2048, and 4096. We compute bits-per-pixel (bpp) as $\log_2 K \times N / H \times W$, where $H, W$ are image height and width. In our case, we cover bpp values of 0.07–1.00. We fix channel dimension to 16 following original Stable Diffusion 3 UNet. For bpp $0.50, 1.00$ we adopt codebook dimension $4, 8$ respectively. For bpp $< 0.50$, we adopt codebook dimension 16. For either UNet or ViT architecture, we follow ReVQ (Zhang et al., 2025) to put codebook dimension on channel dimension. Metrics include Peak Signal-to-Noise Ratio (**PSNR**), Learned Perceptual Image Patch Similarity (**LPIPS**) (Zhang et al., 2018), Structural Similarity Index Measure (**SSIM**) (Wang et al., 2004), and reconstruction Fréchet Inception Distance (**rFID**) (Heusel et al., 2017) for image reconstruction; and generation Fréchet Inception Distance (**gFID**) and Inception Score (**IS**) (Salimans et al., 2016) for image generation. Further details are provided in Appendix C.

## 4.2. Main Results

**Image Reconstruction** Tables 1 and 15 compare GQ with other quantization methods with same model architecture. Across reconstruction metrics such as PSNR, LPIPS, SSIM, and rFID, GQ achieves state-of-the-art performance in most cases. Its advantage is consistent across UNet and ViT architectures, as well as on the ImageNet and COCO datasets. Moreover, the quantization brought by GQ is marginal compared to Gaussian VAE. Figure 2 shows that GQ produces

*Table 2.* Quantitative results on the ImageNet dataset, compared with pre-trained VQ-VAEs. Our GQ outperforms prior pre-trained VQ-VAEs while requiring less training. **Bold**: best, *: from paper, -: not available.

| Method | bpp ($K \times N$) | PSNR↑ | LPIPS↓ | SSIM↑ | rFID↓ | ImageNet Trained Epochs↓ | Params(M)↓ |
|---|---|---|---|---|---|---|---|
| TokenBridge* (Wang et al., 2025) | 0.375 ($2^6 \times 4096$) | - | - | - | 1.11 | - | 83 |
| OpenMagViT-V2* (Luo et al., 2024) | | 21.63 | **0.111** | 0.640 | 1.17 | 300 | 170 |
| ReVQ-256T* (Zhang et al., 2025) | 0.07 ($2^{14} \times 256$) | 21.96 | 0.121 | 0.640 | 2.05 | (Mixed dataset as DC-AE) | 83 |
| GQ (Ours) | | **22.30** | 0.116 | **0.642** | **1.04** | 40 | 87 |
| VQGAN-Taming* (Esser et al., 2021) | | 23.38 | - | - | 1.190 | (OpenImages) | 67 |
| VQGAN-SD* (Rombach et al., 2022) | | - | - | - | 1.140 | (OpenImages) | 83 |
| Llama-Gen-32* (Sun et al., 2024b) | 0.22 ($2^{14} \times 1024$) | 24.44 | **0.064** | 0.768 | 0.590 | 40 | 70 |
| FlowMo-Hi* (Sargent et al., 2025) | | 24.93 | 0.073 | **0.785** | 0.560 | 300 | 945 |
| GQ (Ours) | | **25.31** | **0.064** | 0.762 | **0.491** | 40 | 87 |
| BSQ* (Zhao et al., 2024) | 0.28 ($2^{18} \times 1024$) | 27.78 | 0.063 | **0.817** | 0.990 | 200 | 175 |
| GQ (Ours) | | **27.86** | **0.054** | 0.804 | **0.424** | 40 | 87 |

*Table 3.* Quantitative results of improving TokenBridge on the UNet architecture. Applying TDC significantly enhances the reconstruction quality of TokenBridge.

| Method | bpp ($K \times N$) | ImageNet validation | | | | COCO validation | | | |
|---|---|---|---|---|---|---|---|---|---|
| | | PSNR↑ | LPIPS↓ | SSIM↑ | rFID↓ | PSNR↑ | LPIPS↓ | SSIM↑ | rFID↓ |
| Gaussian VAE | $\approx 1.00$ (-) | 32.73 | 0.022 | 0.910 | 0.490 | 32.64 | 0.018 | 0.917 | 2.380 |
| Gaussian VAE (w/ TDC) | | 32.61 | 0.023 | 0.906 | 0.460 | 32.69 | 0.019 | 0.919 | 2.717 |
| TokenBridge | | 28.24 | 0.045 | 0.869 | 0.823 | 28.19 | 0.043 | 0.878 | 4.167 |
| TokenBridge (w/ TDC) | 1.00 ($2^{16} \times 4096$) | 31.67 | 0.025 | 0.903 | 0.385 | 31.56 | 0.022 | 0.910 | 2.171 |
| GQ (Ours) | | 32.60 | 0.022 | 0.908 | 0.280 | 32.53 | 0.020 | 0.917 | 2.153 |

visually pleasing reconstructions, preserving substantially more details from the source image. Moreover, Table 2 shows that GQ achieves competitive performance relative to several pre-trained models, such as FlowMo, while requiring fewer training epochs and smaller or comparable model size. Finally, GQ outperforms prior Gaussian VAE discretization methods, including TokenBridge and ReVQ.

**Improving TokenBridge** Table 3 compares TokenBridge (Wang et al., 2025) applied to a vanilla Gaussian VAE versus a TDC-constrained Gaussian VAE. The results show that TokenBridge incurs substantial quantization error on a vanilla Gaussian VAE, whereas applying TDC significantly reduces this error.

**Image Generation** Table 6 evaluates GQ for image generation. Compared with VQGAN, FSQ, LFQ, and BSQ, GQ achieves higher codebook usage and codebook entropy. In terms of generation FID and IS, GQ is comparable to FSQ and outperforms the other methods. Additionally, we train a DiT (Peebles & Xie, 2022) with the same architecture and training setup using Gaussian VAEs with and without TDC. The results show that, under limited computational resources, autoregressive generation is more efficient than diffusion in both FID and IS. This demonstrates that converting a Gaussian VAE to a VQ-VAE facilitates autoregressive generation, improving image generation efficiency.

**Complexity** Compared with a Gaussian VAE, the computational overhead of GQ is negligible (see Appendix D.6).

### 4.3. Ablation Study

**GQ Conversion: Alternatives** Stochastic alternatives to GQ conversion exist, including MRC, ORC, and A* coding (Havasi et al., 2018b; Theis & Yosri, 2021; Flamich et al., 2022; He et al., 2024). Table 4 compares these methods in terms of reconstruction quality. When applied to a TDC-constrained Gaussian VAE, GQ achieves the best PSNR, SSIM, and rFID. Moreover, for codebook dimension $m = 1$, GQ can be implemented via bisection search, making it asymptotically faster (see Appendix D.7).

**GQ Conversion: Selection of Codebook Size** Theorem 3.1-3.2 show that setting codebook size $\log K = R_i$ is theoretically optimal. In Table 7, we show that empirically GQ is most effective when $\log K$ matches $R_i$.

**GQ Conversion: Robustness to Random Codebook** The codebook $c_{1:K}$ is sampled from a Gaussian distribution and fixed once generated. Table 8 shows that GQ's performance is nearly invariant to different random seeds.

**GQ Training: TDC** To evaluate the necessity of TDC in Eq. 7, we train a vanilla Gaussian VAE without TDC. As shown in Table 5, the mean $R_i^{\text{bits}}$ of the vanilla VAE is close

*Table 4.* Comparison between GQ conversion and its alternatives on the ImageNet dataset. GQ achieves better reconstruction quality and can be implemented asymptotically faster when the codebook dimension $m = 1$. Here, $D_\infty(\cdot||\cdot)$ denotes the Rényi $\infty$-divergence.

| Methods | Encoding / Decoding Complexity | PSNR↑ | LPIPS↓ | SSIM↑ | rFID↓ |
|---|---|---|---|---|---|
| Gaussian VAE (w/ TDC) | $O(1)/O(1)$ | 32.75 | 0.023 | 0.906 | 0.460 |
| MRC (original) | $O(2^{R_i})/O(1)$ | 32.09 | 0.023 | 0.906 | 0.425 |
| MRC (A* coding) | $O(D_\infty(q(Z_i|X)||\mathcal{N}(0,1)))/O(D_\infty(q(Z_i|X)||\mathcal{N}(0,1)))$ | 32.09 | 0.023 | 0.906 | 0.425 |
| ORC | $O(2^{R_i})/O(1)$ | 32.09 | 0.023 | 0.906 | 0.419 |
| GQ (Ours) | $O(R_i)/O(1)$ | 32.11 | 0.023 | 0.907 | 0.414 |

*Table 5.* Effect of TDC and alternatives. GQ is effective only when applied to a Gaussian VAE trained with the TDC constraint.

| Methods | Constraint | $R_i^{\text{bits}}$ mean, min-max | $\log_2 K$ | bpp | PSNR↑ | LPIPS↓ | SSIM↑ | rFID↓ |
|---|---|---|---|---|---|---|---|---|
| Gaussian VAE | None | 3.99, 0.26-27.29 | - | 1.00 | 32.73 | 0.022 | 0.910 | 0.490 |
| GQ | - | - | 4 | 1.00 | 26.43 | 0.054 | 0.834 | 0.978 |
| Gaussian VAE | MIRACLE / HiFiC | 4.34, 0.91-26.98 | - | 1.00 | 32.82 | 0.023 | 0.910 | 0.436 |
| GQ | - | - | 4 | 1.00 | 29.48 | 0.039 | 0.887 | 0.439 |
| Gaussian VAE | IsoKL | 4.34, 4.24-4.38 | - | 1.00 | 30.54 | 0.878 | 0.027 | 0.400 |
| GQ | - | - | 4 | 1.00 | 30.45 | 0.878 | 0.030 | 0.468 |
| Gaussian VAE | TDC (Ours, m=1) | 4.26, 2.93-5.63 | - | 1.06 | 32.61 | 0.023 | 0.906 | 0.460 |
| GQ | - | - | 4 | 1.00 | 32.11 | 0.023 | 0.906 | 0.414 |

*Table 6.* Quantitative results for class-conditional image generation on the ImageNet dataset. GQ achieves the highest codebook utilization while maintaining competitive generation performance. **Bold**: best. Underline: second best.

| Method | Codebook usage↑ | gFID ↓ | IS ↑ |
|---|---|---|---|
| *Diffusion* | | | |
| Gaussian VAE w/o TDC | - | 8.35 | 202.19 |
| Gaussian VAE w/ TDC | - | 8.47 | 205.94 |
| *Auto-regressive* | | | |
| VQGAN | 16.4% | 8.01 | 151.40 |
| FSQ | 94.3% | **7.33** | 224.88 |
| LFQ | 24.9% | 7.73 | 142.09 |
| BSQ | 99.8% | 7.82 | 221.64 |
| TokenBridge | 94.6% | 7.82 | 198.24 |
| GQ (Ours) | **100.0%** | 7.67 | **230.79** |

*Table 8.* GQ is robust to random codebook.

| Random Seed | PSNR↑ | LPIPS↓ | SSIM↑ | rFID↓ |
|---|---|---|---|---|
| 42 | 27.61 | 0.059 | 0.807 | 0.529 |
| 43 | 27.61 | 0.059 | 0.807 | 0.523 |
| 44 | 27.62 | 0.059 | 0.807 | 0.526 |

*Table 9.* The necessity of GQ's two-stage pipeline is demonstrated: converting a pre-trained Gaussian VAE yields better performance than training GQ from scratch, and further fine-tuning provides only marginal improvements.

| Method | bpp | PSNR↑ | LPIPS↓ | SSIM↑ | rFID↓ |
|---|---|---|---|---|---|
| VQ-VAE from scratch | | 8.50 | 0.763 | 0.156 | 360 |
| Eq. 8 from scratch | | 29.65 | 0.044 | 0.866 | 0.928 |
| Convert from Eq. 8 | 1.00 | 32.47 | 0.023 | 0.907 | 0.327 |
| Further finetune Eq. 8 | | 32.45 | 0.022 | 0.905 | 0.264 |

to that of the TDC-constrained VAE (3.99 vs. 4.26 bits), but the range of $R_i^{\text{bits}}$ is much wider for the vanilla model (0.26–27.29 vs. 2.93–5.63 bits). While the reconstruction performance of the two Gaussian VAEs is similar (PSNR: 32.73 vs. 32.61 dB, rFID: 0.490 vs. 0.460), GQ applied to the TDC-constrained VAE significantly outperforms GQ

*Table 7.* The reconstruction result when $\log K \neq R_i$. GQ works best when $\log K$ matches $R_i$.

| $R_i^{\text{bits}}$ | $\log_2 K$ | $K \times N$ | PSNR↑ | LPIPS↓ | SSIM↑ | rFID↓ |
|---|---|---|---|---|---|---|
| 14 | 14 | $2^{14} \times 1024$ | 25.31 | 0.064 | 0.762 | 0.491 |
| 18 | 14 | $2^{14} \times 1024$ | 25.24 | 0.068 | 0.774 | 0.527 |
| 14 | 18 | $2^{18} \times 1024$ | 27.79 | 0.059 | 0.808 | 0.513 |
| 18 | 18 | $2^{18} \times 1024$ | 27.86 | 0.054 | 0.804 | 0.424 |

applied to the vanilla VAE (PSNR: 31.25 vs. 26.43 dB, rFID: 0.372 vs. 0.978). This demonstrates that TDC is essential for effective GQ conversion.

Alternatives to TDC include the MIRACLE / HiFiC heuristic (Havasi et al., 2018a; Mentzer et al., 2020) and IsoKL (Flamich et al., 2022). MIRACLE / HiFiC is less effective at controlling the range and is outperformed by TDC (PSNR 29.48 vs. 32.11 dB). IsoKL imposes a stricter constraint by requiring that $R_i$ is identical across all dimensions. While IsoKL enforces this constraint effectively, its overall performance is worse (PSNR 30.45 vs. 32.11 dB) due to numerical instability and the exclusion of solutions with $\sigma_i^2 > 1$. In Appendix B, we propose a numerically stable version of Mean-KL (Lin et al., 2023), which extends IsoKL

*Table 10.* Ablation study on hyperparameters $\alpha$, $\beta$, and codebook dimension $m$. Here, $\alpha^{\text{bits}} = \alpha / \log 2$.

| $\alpha^{bits}$ | $\beta$ | $m$ | PSNR↑ | LPIPS↓ | SSIM↑ | rFID↓ |
|---|---|---|---|---|---|---|
| 0.5 | 1.01 | 1 | 25.60 | 0.088 | 0.702 | 1.26 |
| 0.5 | 1.01 | 4 | 26.98 | 0.068 | 0.793 | 0.927 |
| 0.5 | 1.01 | 16 | 27.61 | 0.059 | 0.807 | 0.529 |
| 0.1 | 1.01 | 16 | 27.56 | 0.058 | 0.812 | 0.551 |
| 1.0 | 1.01 | 16 | 27.61 | 0.063 | 0.811 | 0.701 |
| 0.5 | 1.1 | 16 | 27.63 | 0.060 | 0.809 | 0.534 |
| 0.5 | 1.001 | 16 | 27.48 | 0.058 | 0.804 | 0.510 |

to support multi-dimensional codebooks ($m > 1$). However, it performs poorly for ViT-based models.

**Necessity of GQ's Two-Stage Pipeline** One could train a vanilla VQ-VAE (Van Den Oord et al., 2017) using the same codebook as GQ, effectively creating a VQ-VAE with a fixed Gaussian noise codebook. Alternatively, the Gaussian VAE network could be trained directly with the GQ target in Eq. 8 using Gumbel-Softmax (Jang et al., 2016; Maddison et al., 2016). However, as shown in Table 9, both approaches fail to converge reliably. Moreover, fine-tuning GQ after initializing with a pre-trained Gaussian VAE yields only marginal performance improvements.

**Hyperparameters** To justify our choice of TDC parameters $\alpha = 0.5$ and $\beta = 1.01$, we perform a grid search with results shown in Table 10. The results indicate that the effect of $\alpha$ and $\beta$ on TDC is minimal as long as $\alpha \leq 0.5$. Table 10 also shows that a larger codebook dimension $m$ improves reconstruction. Accordingly, we set $m$ to the quotient of the total latent dimension and the total number of tokens, which is the maximum feasible value. To achieve optimal reconstruction, we use $\omega = 2.0$ for bitrates $\leq 0.50$ bpp and $\omega = 0.0$ for 1.00 bpp (see Table 14 for details).

## 5. Related works

**Vector-Quantized Variational Autoencoder** VQ-VAE (Van Den Oord et al., 2017) is an autoencoder that compresses images into discrete tokens. Due to discretization, it cannot be trained directly via gradient descent. Various techniques address this, including commitment loss (Van Den Oord et al., 2017), expectation maximization (EM) (Roy et al., 2018), the straight-through estimator (STE) (Bengio et al., 2013), and Gumbel-Softmax (Jang et al., 2016; Maddison et al., 2016; Sønderby et al., 2017; Shi et al., 2024b). VQ-VAE is also prone to codebook collapse. To mitigate this, methods such as reducing code dimension (Yu et al., 2021a; Sun et al., 2024a), product quantization (Zheng et al., 2022), residual quantization (Lee et al., 2022), dynamic quantization (Huang et al., 2023), multi-level quantization (Razavi et al., 2019), feature projection (Zhu et al., 2024), and rotation codebooks (Fifty et al., 2024) have been

proposed (Yu et al., 2021b; Chiu et al., 2022; Takida et al., 2022; Zhang et al., 2023; Huh et al., 2023; Gautam et al., 2023; Goswami et al., 2024).

More closely related to our work, some VQ-VAE variants have fixed codebooks, including FSQ (Mentzer et al., 2023), LFQ (Yu et al., 2023), BSQ (Zhao et al., 2024) and their extensions (Zhuang et al., 2025; Lin et al., 2026). However, training still relies on tricks such as the straight-through estimator (STE). On the other hand, TokenBridge (Wang et al., 2025) and ReVQ (Zhang et al., 2025) convert a pre-trained Gaussian VAE into a VQ-VAE, but they do not constrain the divergence of the Gaussian VAE.

**Reverse Channel Coding** GQ is closely related to reverse channel coding, which aims to simulate a distribution $q$ using samples from another distribution $p$ (Harsha et al., 2007; Li & El Gamal, 2018; Havasi et al., 2018b; Flamich et al., 2020; Theis & Yosri, 2021; Flamich et al., 2022; He et al., 2024). The key difference between MRC and GQ is that MRC and its variants (Havasi et al., 2018b; Theis & Yosri, 2021; Flamich et al., 2022; He et al., 2024) simulate a distribution via stochastic sampling, whereas a VQ-VAE requires deterministic quantization. Regarding quantization error, GQ outperforms MRC by construction (Eq. 3). Moreover, GQ with a one-dimensional codebook ($m = 1$) can be implemented via bisection search, achieving superior asymptotic complexity (see Appendix D.7).

Additionally, TDC is closely related to the MIRACLE / HiFiC heuristic and IsoKL parameterization of Gaussian VAEs (Havasi et al., 2018a; Mentzer et al., 2020; Flamich et al., 2022; Lin et al., 2023). Specifically, MIRACLE / HiFiC also adjusts $\lambda$ during VAE training, but it maintains a single $\lambda$, making it less effective at controlling the minimum and maximum values of $R_i$. IsoKL, in contrast, enforces strict control over $R_i$ by directly solving for $\sigma$ given $\mu$ using the Lambert $\mathcal{W}$ function (Corless et al., 1996; Brezinski, 1996), but it suffers from numerical instability and yields suboptimal performance.

## 6. Conclusion & Discussion

In this paper, we propose **Gaussian Quant (GQ)**, a simple yet effective method that constructs a VQ-VAE by first training a constrained Gaussian VAE then converting it into a VQ-VAE. The conversion process involves generating codebook from pure Gaussian noise and matching the posterior mean. Theoretically, we show that when the logarithm of the GQ codebook size exceeds the bits-back coding bitrate of the Gaussian VAE, the resulting quantization error is small. To train a constrained Gaussian VAE for effective conversion, we introduce the target divergence constraint (TDC). Empirically, GQ outperforms prior discrete VAEs, including VQGAN, FSQ, LFQ, and BSQ (Van Den Oord

et al., 2017; Mentzer et al., 2023; Yu et al., 2023; Zhao et al., 2024), and TDC further improves the performance of TokenBridge (Wang et al., 2025).

One disadvantage of GQ is that it is more complex to implement and contains additional hyper-parameter on top of VQ. However, it is noteworthy that GQ also eliminates the need of STE, codebook loss, commitment loss and entropy loss. We acknowledge that several highly competitive VQ-VAEs employ multi-scale or residual architectures (Razavi et al., 2019; Lee et al., 2022; Tian et al., 2024; Han et al., 2024). In this work, however, we use a standard single-scale architecture to focus on the core aspects of the quantization method. Furthermore, recent studies show that strong reconstruction does not necessarily imply strong generation, and achieving good generation often requires feature alignment (Wang et al., 2024; Xiong et al., 2025; Hansen-Estruch et al., 2025). Here, we focus on reconstruction and leave the exploration of the complex relationship between reconstruction and generation performance to future work.

## Impact Statement

The approach proposed in this paper focus on reconstruction of existing images with limited bitrate. As the model is essentially not generative, the ethic concerns is not obvious. Nevertheless, the GAN module in decoder might has negative effects, including issues related to mis-representation and trustworthiness.

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

## A. Proof of Main Results

**Lemma A.1.** *Given $R_i = D_{KL}(q(Z_i|X)||\mathcal{N}(0,1)), |\mu_i|, |\sigma_i|$ are upper-bounded. And therefore $c_1, c_2$ in Theorem 3.1-3.2 are also bounded:*

$$|\mu_i| \leq \sqrt{2R_i},$$

$$|\sigma_i| \leq \sqrt{-\mathcal{W}_{-1}(-e^{-(2R_i+1)})},$$

$$c_1 \leq \sqrt{2R_i}\sqrt{-\mathcal{W}_{-1}(-e^{-(2R_i+1)})},$$

$$c_2 \leq \sqrt{2R_i} + \sqrt{-\mathcal{W}_{-1}(-e^{-(2R_i+1)})}.$$

*Proof.* First, we have

$$\mu_i^2 + \sigma_i^2 - 1 - \log \sigma_i^2 = 2R_i. \tag{10}$$

Notice that $\sigma_i^2 - \log \sigma_i^2 \geq 0$. To maximize $\sigma_i$, we simply set $\mu_i = 0$. Then we have $\sigma_i^2 - \log \sigma_i^2 = 2R_i + 1$. We can solve

$$\sigma_i^2 = -\mathcal{W}(e^{-(2R_i+1)}), \tag{11}$$

where $\mathcal{W}(.)$ is Lambert $\mathcal{W}$ function. We select the larger solution

$$\sigma^2 = -\mathcal{W}_{-1}(e^{-(2R_i+1)}), \tag{12}$$

where $\mathcal{W}_{-1}$ is the lower branch of Lambert $\mathcal{W}$ function. To maximize $\mu_i$, we simply set $\sigma_i = 1$. Then we have

$$\mu_i^2 = 2R_i, \mu_i = \sqrt{2R_i}. \tag{13}$$

$\square$

**Theorem 3.1.** *Denote the mean and standard deviation of $q(Z_i|X = x)$ as $\mu_i$ and $\sigma_i$, respectively. We assume $|\mu_i\sigma_i| \leq c_1$ and $|\mu_i| + |\sigma_i| \leq c_2$. Given a fixed $R_i = D_{KL}(q(Z_i|X)||\mathcal{N}(0,1))$, the probability of a quantization error $|\hat{z}_i - \mu_i| \geq \sigma_i$ decays doubly exponentially with the number of nats $t$ by which the codebook bitrate $\log K$ exceeds the bits-back coding rate. That is,*

$$\text{when } \log K = R_i + t,$$

$$\Pr\{|\hat{z}_i - \mu_i| \geq \sigma_i\} \leq \exp\left(-e^t\sqrt{\frac{2}{\pi}}e^{-c_1-0.5}\right). \tag{14}$$

*Proof.* Denote the cumulative distribution function (CDF) of $\mathcal{N}(0,1)$ as $\Phi$, and probability density function (PDF) of $\mathcal{N}(0,1)$ as $\phi$, then we need to consider the probability that no samples falls between $[\mu_i - \sigma_i, \mu_i + \sigma_i]$, which is

$$\Pr\{|\hat{z}_i - \mu_i| \geq \sigma_i\} = (1 - (\Phi(\mu_i + \sigma_i) - \Phi(\mu_i - \sigma_i)))^K. \tag{15}$$

Now we use the Bernoulli inequality, that $\forall y \in \mathbb{R}, 1 + y \leq e^y$. Let $y = -(\Phi(\mu_i + \sigma_i) - \Phi(\mu_i - \sigma_i))$, we have

$$1 - (\Phi(\mu_i + \sigma_i) - \Phi(\mu_i - \sigma_i)) \leq \exp\left(-(\Phi(\mu_i + \sigma_i) - \Phi(\mu_i - \sigma_i))\right). \tag{16}$$

Taking Eq. 16 into Eq. 15, we have

$$\begin{aligned}
\Pr\{|\hat{z}_i - \mu_i| \geq \sigma_i\} &\leq \exp\left(-(\Phi(\mu_i + \sigma_i) - \Phi(\mu_i - \sigma_i))\right)^K \\
&= \exp\left(-K \cdot (\Phi(\mu_i + \sigma_i) - \Phi(\mu_i - \sigma_i))\right) \\
&= \exp\left(-K \cdot \int_{\mu_i-\sigma_i}^{\mu_i+\sigma_i} \phi(x)dx\right).
\end{aligned} \tag{17}$$

By integral mean value theorem, $\exists x' \in [\mu_i - \sigma_i, \mu_i + \sigma_i]$, such that

$$\int_{\mu_i - \sigma_i}^{\mu_i + \sigma_i} \phi(x) dx = 2\sigma_i \phi(x'). \tag{18}$$

And then we have

$$\Pr\{|\hat{z}_i - \mu_i| \geq \sigma_i\} \leq \exp\left(-K \cdot 2\sigma_i \phi(x')\right). \tag{19}$$

Next, we consider three cases: $\mu_i - \sigma_i \geq 0$, $\mu_i + \sigma_i \leq 0$, and $\mu_i - \sigma_i \leq 0 \leq \mu_i + \sigma_i$.

First, consider the case when $\mu_i - \sigma_i \geq 0$. Obviously we have $\phi(\mu_i + \sigma_i) \leq \phi(x')$, and we have

$$
\begin{aligned}
\Pr\{|\hat{z}_i - \mu_i| \geq \sigma_i\} &\leq \exp\left(-K \cdot 2\sigma_i \phi(\mu_i + \sigma_i)\right) \\
&= \exp\left(-K \cdot \sqrt{\frac{2}{\pi}} \sigma_i e^{-\frac{1}{2}(\mu_i + \sigma_i)^2}\right) \\
&= \exp\left(-K \cdot \sqrt{\frac{2}{\pi}} e^{-\frac{1}{2}(\mu_i^2 + \sigma_i^2 - \log \sigma^2 - 1.0 + 1.0) - \mu_i \sigma_i}\right) \\
&= \exp\left(-K \cdot \sqrt{\frac{2}{\pi}} e^{-R_i - \mu_i \sigma_i - 0.5}\right).
\end{aligned}
\tag{20}
$$

Notice that as $\mu_i - \sigma_i \geq 0$, $\sigma_i > 0$, we must have $\mu_i \sigma_i > 0$, then

$$\Pr\{|\hat{z}_i - \mu_i| \geq \sigma_i\} \leq \exp\left(-K \cdot \sqrt{\frac{2}{\pi}} e^{-R_i - |\mu_{\max} \sigma_{\max}| - 0.5}\right). \tag{21}$$

Similarly, we can show similar result for $\mu_i + \sigma_i \leq 0$. Obviously we have $\phi(\mu_i - \sigma_i) \leq \phi(x')$, and we have

$$
\begin{aligned}
\Pr\{|\hat{z}_i - \mu_i| \geq \sigma_i\} &\leq \exp\left(-K \cdot 2\sigma_i \phi(\mu_i - \sigma_i)\right) \\
&= \exp\left(-K \cdot \sqrt{\frac{2}{\pi}} \sigma_i e^{-\frac{1}{2}(\mu_i + \sigma_i)^2}\right) \\
&= \exp\left(-K \cdot \sqrt{\frac{2}{\pi}} e^{-\frac{1}{2}(\mu_i^2 + \sigma_i^2 - \log \sigma^2 - 1.0 + 1.0) + \mu_i \sigma_i}\right) \\
&= \exp\left(-K \cdot \sqrt{\frac{2}{\pi}} e^{-R_i + \mu_i \sigma_i - 0.5}\right) \\
&\leq \exp\left(-K \cdot \sqrt{\frac{2}{\pi}} e^{-R_i - |\mu_{\max} \sigma_{\max}| - 0.5}\right)
\end{aligned}
\tag{22}
$$

Now, consider the case when $\mu_i - \sigma_i < 0 < \mu_i + \sigma_i$, obviously we must have either $\phi(\mu_i - \sigma_i) \leq \phi(x')$, or $\phi(\mu_i + \sigma_i) \leq \phi(x')$. If $\phi(\mu_i + \sigma_i) \leq \phi(x')$, then the result is the same as $\mu - \sigma_i \geq 0$. If $\phi(\mu_i - \sigma_i) \leq \phi(x')$, then the result is the same as $\mu + \sigma_i \leq 0$.

Therefore, for all $\mu_i, \sigma_i$, we have

$$\Pr\{|\hat{z}_i - \mu_i| \geq \sigma_i\} \leq \exp\left(-K \cdot \sqrt{\frac{2}{\pi}} e^{-R_i - |\mu_{\max} \sigma_{\max}| - 0.5}\right). \tag{23}$$

Taking the value of $K$ in, we have the result

$$
\begin{aligned}
\Pr\{|\hat{z}_i - \mu_i| \geq \sigma_i\} &\leq \exp\left(-K \cdot \sqrt{\frac{2}{\pi}} e^{-R_i - |\mu_{\max} \sigma_{\max}| - 0.5}\right) \\
&= \exp\left(-e^t \cdot \sqrt{\frac{2}{\pi}} e^{-|\mu_{\max} \sigma_{\max}| - 0.5}\right) \\
&= \exp\left(-e^t \cdot \sqrt{\frac{2}{\pi}} e^{-c_1 - 0.5}\right).
\end{aligned}
\tag{24}
$$

$\square$

**Theorem 3.2.** *Following Theorem [3.1], the probability of a quantization error $|\hat{z}_i - \mu_i| \geq \sigma_i$ increases exponentially with the number of nats $t$ by which the codebook bitrate $\log K$ falls below the bits-back coding rate. That is,*

$$\text{when } \log K = R_i - t,$$

$$\Pr\{|\hat{z}_i - \mu_i| \geq \sigma_i\} \geq 1 - e^{-t}\sqrt{\frac{2}{\pi}}e^{0.5c_2^2 - 0.5}. \tag{25}$$

*Proof.* Similar to the proof of Theorem. [3.1], we have

$$\Pr(|\hat{z}_i - \mu_i| \geq \sigma_i) = (1 - (\Phi(\mu_i + \sigma_i) - \Phi(\mu_i - \sigma_i)))^K. \tag{26}$$

Now we use an inequality, that $\forall y \in (0,1), K \in \mathbb{N}, K \geq 1, (1-y)^K \geq 1 - Ky$. This is due to the fact that $(1-y)^K$ is convex in $(0,1)$, and $1 - Ky$ is tangent line at $y = 0$. With this inequality, we have

$$(1 - (\Phi(\mu_i + \sigma_i) - \Phi(\mu_i - \sigma_i)))^K \geq 1 - K(\Phi(\mu_i + \sigma_i) - \Phi(\mu_i - \sigma_i)). \tag{27}$$

Again, we can use integral mean value theorem, and find out that when $\mu_i - \sigma_i \geq 0$,

$$\begin{aligned}
\Pr(|\hat{z}_i - \mu_i| \geq \sigma_i) &\geq 1 - K(\Phi(\mu_i + \sigma_i) - \Phi(\mu_i - \sigma_i)) \\
&\geq 1 - K2\sigma_i\phi(\mu_i - \sigma_i) \\
&= 1 - K\sqrt{\frac{2}{\pi}}\sigma_i e^{-\frac{1}{2}(x_i - \sigma_i)^2} \\
&= 1 - K\sqrt{\frac{2}{\pi}}e^{-\frac{1}{2}(x_i^2 + \sigma_i^2 - \log\sigma_i^2 - 1.0) + |\mu_i\sigma_i| - 0.5} \\
&= 1 - K\sqrt{\frac{2}{\pi}}e^{-R_i + |\mu_i\sigma_i| - 0.5} \\
&\geq 1 - Ke^{-R_i}\sqrt{\frac{2}{\pi}}e^{0.5(\mu_i + \sigma_i)^2 - 0.5}
\end{aligned} \tag{28}$$

Similar results can be obtained for $\mu_i + \sigma_i \leq 0$. For the case that $\mu_i - \sigma_i \leq 0 \leq \mu_i + \sigma_i$, we have

$$\begin{aligned}
\Pr(|\hat{z}_i - \mu_i| \geq \sigma_i) &\geq 1 - K(\Phi(\mu_i + \sigma_i) - \Phi(\mu_i - \sigma_i)) \\
&\geq 1 - K2\sigma_i\phi(0) \\
&= 1 - K\sqrt{\frac{2}{\pi}}\sigma_i e^{-\frac{1}{2}(0)^2} \\
&= 1 - K\sqrt{\frac{2}{\pi}}e^{-\frac{1}{2}(\mu_i^2 + \sigma_i^2 - \log\sigma_i^2 - 1.0) - 0.5 + 0.5(\mu_i^2 + \sigma_i^2)} \\
&= 1 - K\sqrt{\frac{2}{\pi}}e^{-R_i + 0.5(\mu_i^2 + \sigma_i^2) + |\mu_i\sigma_i| - 0.5} \\
&= 1 - Ke^{-R_i}\sqrt{\frac{2}{\pi}}e^{0.5(\mu_i + \sigma_i)^2 - 0.5}
\end{aligned} \tag{29}$$

Taking the value of $K = e^{R_i - t}$, we have the result

$$\begin{aligned}
\Pr(|\hat{z}_i - \mu_i| \geq \sigma_i) &\geq 1 - e^{R_i - t - R_i}\sqrt{\frac{2}{\pi}}e^{0.5(\mu_i + \sigma_i)^2 - 0.5} \\
&\geq 1 - e^{-t}\sqrt{\frac{2}{\pi}}e^{0.5(|\mu_i| + |\sigma_i|)^2 - 0.5} \\
&\geq 1 - e^{-t}\sqrt{\frac{2}{\pi}}e^{0.5c_2^2 - 0.5}.
\end{aligned} \tag{30}$$

$\square$

## B. Stable Mean-KL Parametrization

We investigate an alternative to TDC, namely the Mean-KL parametrization (Lin et al., 2023), which is considered to be easier to train than TDC since it does not require the construction of an empirical $R(\lambda)$ model.

### B.1. Mean-KL Parametrization

The Mean-KL parametrization (Lin et al., 2023) supports codebook dimension $m > 1$. Its neural network output consists of two $m$-dimensional tensors, $\hat{\gamma}_{i:i+m}$ and $\tau_{i:i+m}$, which allocate the $R_{i:i+m}$ target $\log K$ across the $m$ dimensions and determine the mean, respectively. More specifically, the Mean-KL parametrization determines the mean $\mu_{i:i+m}$ and variance $\sigma^2_{i:i+m}$ as follows, where $\mathcal{W}(\cdot)$ denotes the principal branch of the Lambert $\mathcal{W}$ function:

$$
\begin{aligned}
\gamma_{i:i+m} &= \text{Softmax}(\hat{\gamma}_{i:i+m}), \\
\kappa_{i:i+m} &= \gamma_{i:i+m} K, \\
\mu_{1:m} &= \sqrt{2\kappa_{i:i+m}}\tanh(\tau_{i:i+m}), \\
\sigma^2_{i:i+m} &= -\mathcal{W}(-\exp(\mu^2_{i:i+m} - 2\kappa_{i:i+m} - 1.0)).
\end{aligned}
\tag{31}
$$

The Mean-KL parametrization is designed for model compression. When directly applied to Gaussian VAEs, two typical cases may arise, as shown in Table 11, both of which can result in a not-a-number (NaN) error in floating-point computations.

*Table 11.* Two typical types of NaN in Mean-KL parametrization.

| $\mu_i$ | $\kappa_i$ | $\sigma^2_i$ |
|---|---|---|
| -2.7286 | 3.7227 | NaN |
| 0.0013 | $9.1458 \times 10^{-7}$ | NaN |

### B.2. Stable Mean-KL Parametrization

It is evident that the two types of NaN errors are caused by excessively large values of $|\mu_i|$ and excessively small values of $\kappa_i$, respectively. To address this numerical issue, we propose the Stable Mean-KL parametrization, which introduces two regularization parameters, $r_1 = 0.1$ and $r_2 = 0.01$. The parameter $r_1$ ensures that each $\kappa_i \geq r_1/m$, while $r_2$ shrinks $\mu_i$ towards 0.

$$
\begin{aligned}
\kappa_{i:i+m} &= \gamma_{i:i+m}(K - r_1) + r_1/m, \\
\mu_{1:m} &= \sqrt{2\kappa_{i:i+m}}\tanh(\tau_{i:i+m})(1 - r_2),
\end{aligned}
\tag{32}
$$

### B.3. Results of Stable Mean-KL Parametrization

In Table 12, we present the results of the Stable Mean-KL parametrization. For UNet-based models, Stable Mean-KL achieves performance comparable to TDC. However, for ViT-based models, Stable Mean-KL performs significantly worse than TDC. Since Stable Mean-KL does not consistently outperform TDC, we choose to use TDC for the final model. Nonetheless, if only UNet-based models are required, Stable Mean-KL can be an effective alternative to TDC, as it does not require an empirical $R(\lambda)$ model and is significantly simpler to train.

*Table 12.* Quantitative results on ImageNet validation dataset comparing GQ with different constraint.

| Method | bpp ($K \times N$) | UNet based | | | | ViT based | | | |
|---|---|---|---|---|---|---|---|---|---|
| | | PSNR↑ | LPIPS↓ | SSIM↑ | rFID↓ | PSNR↑ | LPIPS↓ | SSIM↑ | rFID↓ |
| GQ (Mean-KL) | | NaN | NaN | NaN | NaN | NaN | NaN | NaN | NaN |
| GQ (Stable Mean-KL) | 1.00 ($2^{16} \times 4096$) | 32.35 | 0.023 | 0.905 | 0.280 | 30.80 | 0.030 | 0.891 | 0.556 |
| GQ (TDC) | | 32.47 | 0.023 | 0.907 | 0.322 | 31.71 | 0.024 | 0.903 | 0.349 |

## C. Implementation Details

### C.1. Details of Training and Distortion Objective

We train all VQ-VAEs on the ImageNet validation dataset using $8\times$ H100 GPUs for approximately 24 hours. For UNet models, we train each model for 30 epochs using the Adam (Kingma & Ba, 2014) optimizer with a batch size of 128 and a learning rate of $1 \times 10^{-4}$. For ViT models, we train each model for 40 epochs using the Adam optimizer with a batch size of 256 and a learning rate of $4 \times 10^{-7}$.

All VQ-VAEs are trained using the following distortion objective, which corresponds to the classical VQ-GAN (Esser et al., 2020) objective employed in the Stable Diffusion VAE (Rombach et al., 2022).

$$\Delta(X, g(z)) = \mathcal{L}_{MSE}(X, g(z)) + w_1\mathcal{L}_{LPIPS}(X, g(z)) + w_2\mathcal{L}_{GAN}(g(z)). \tag{33}$$

Following the implementation of Stable Diffusion, we set $w_1 = 1.0$ and $w_2 = 0.75$ for UNet models. Consistent with the implementation of BSQ (Zhao et al., 2024), we set $w_1 = 0.1$ and $w_2 = 0.1$ for ViT models.

For the image generation model, we first train all VQ-VAEs using images of size $128 \times 128$, following the same settings as described above. Subsequently, we train the auto-regressive transformer for image generation using the implementation of IBQ (Shi et al., 2024b) with a Llama-base transformer architecture. The transformer has a vocabulary size of $2^{16}$, 16 layers, 16 attention heads, and an embedding dimension of 1024. We train the transformer for 100 epochs using the Adam optimizer with a learning rate of $3 \times 10^{-4}$ and a batch size of 512.

### C.2. Details of Hyper-parameters

Below, we describe the implementation details along with the definition of hyperparameters for each method. In Table 13, we list the values of these hyperparameters for different bits-per-pixel (bpp) settings.

**VQGAN** (Van Den Oord et al., 2017) We adopt the factorized codebook VQGAN variant following (Zheng et al., 2022). For each codebook, we use a codebook size of $K = 2^{16}$ and a group dimension of $m = 16$. The number of codebooks $n$ varies depending on the bitrate. Additionally, we use a codebook loss weight of $\lambda = 1.0$ and a commitment loss weight of $\zeta = 0.25$.

**FSQ** (Mentzer et al., 2023) The only parameter of FSQ is the codebook list $l$, which represents the quantization level for each integer value. We set each unit value to $2^4 = 16$, and populate $l$ with the appropriate number of 16s according to the desired bitrate.

**LFQ** (Yu et al., 2023) For LFQ at 0.25 bpp, we follow the original paper and split a large codebook of size $2^{16}$ into $n = 2$ smaller codebooks, each with $K = 2^8$ and a codebook dimension of $m = 8$. We use an entropy loss weight of $\lambda = 0.1$ and a commitment loss weight of $\zeta = 0.025$.

**BSQ** (Zhao et al., 2024) We fix the size of each BSQ codebook to $2^1$, with a group dimension of $m = 1$, and vary the number of codebooks $n$ according to the desired bitrate. For the entropy penalization parameter, we set $\lambda = 0.1$, following the official implementation.

**GQ** We use a fixed codebook size of $K = 2^{16}$. We adopt fixed $\alpha = 0.5, \beta = 1.01$. We adopt $m = 16, 8, 4, \omega = 2.0, 2.0, 0.0$ for bpp= $0.25, 0.5, 1.0$ respectively.

## D. Additional Quantitative Results

### D.1. Detail in Codebook Usage Hyper-parameter $\omega$

Additionally, in Table 14, we show the effect of the codebook usage regularization parameter $\omega$. For high bitrates, such as 1.00 bpp, regularization is not required; in other words, setting $\omega = 0.0$ yields the good enough codebook usage and rFID. For lower bitrates, such as 0.50 bpp, $\omega = 0.0$ leads to codebook collapse, while $\omega = 2.0$ achieves the best codebook entropy and rFID.

Table 13. Details of Hyper-parameter values.

| | bpp | Hyper-parameters |
|---|---|---|
| | 0.25 | $K = 2^{16}, n = 1, m = 16, \lambda = 1.0, \zeta = 0.25$ |
| VQ | 0.50 | $K = 2^{16}, n = 2, m = 16, \lambda = 1.0, \zeta = 0.25$ |
| | 1.00 | $K = 2^{16}, n = 4, m = 16, \lambda = 1.0, \zeta = 0.25$ |
| | 0.25 | $l = \{16, 16, 16, 16\}$ |
| FSQ | 0.50 | $l = \{16, 16, 16, 16, 16, 16, 16, 16\}$ |
| | 1.00 | $l = \{16, 16, 16, 16, 16, 16, 16, 16, 16, 16, 16, 16, 16, 16, 16, 16\}$ |
| | 0.25 | $K = 2^8, n = 2, m = 8, \lambda = 0.1, \zeta = 0.025$ |
| LFQ | 0.50 | $K = 2^8, n = 4, m = 8, \lambda = 0.1, \zeta = 0.025$ |
| | 1.00 | $K = 2^8, n = 8, m = 8, \lambda = 0.1, \zeta = 0.025$ |
| | 0.25 | $K = 2^1, n = 16, m = 1, \lambda = 0.1$ |
| BSQ | 0.50 | $K = 2^1, n = 32, m = 1, \lambda = 0.1$ |
| | 1.00 | $K = 2^1, n = 64, m = 1, \lambda = 0.1$ |
| | 0.25 | $K = 2^{16}, n = 1, m = 16, \omega = 2.0$ |
| GQ | 0.50 | $K = 2^{16}, n = 2, m = 8, \omega = 2.0$ |
| | 1.00 | $K = 2^{16}, n = 4, m = 4, \omega = 0.0$ |

Table 14. Ablation Study on regularization $\omega$.

| bpp | $\omega$ | Codebook Usage↑ | Codebook Entropy↑ | PSNR↑ | LPIPS↓ | SSIM↑ | rFID↓ |
|---|---|---|---|---|---|---|---|
| | 0.0 | 99.3% | 14.96 | 30.00 | 0.044 | 0.873 | 0.783 |
| | 1.0 | 100.0% | 15.14 | 30.35 | 0.040 | 0.877 | 0.589 |
| 0.50 | 2.0 | 100.0% | 15.22 | 30.17 | 0.039 | 0.875 | 0.492 |
| | 4.0 | 100.0% | 14.81 | 28.08 | 0.061 | 0.846 | 1.269 |
| | 0.0 | 100.0% | 15.05 | 32.47 | 0.023 | 0.907 | 0.322 |
| | 1.0 | 100.0% | 15.05 | 32.47 | 0.023 | 0.907 | 0.327 |
| 1.00 | 2.0 | 100.0% | 15.06 | 32.47 | 0.024 | 0.907 | 0.332 |
| | 4.0 | 100.0% | 15.07 | 32.44 | 0.024 | 0.907 | 0.343 |

## D.2. The Quantization Error in Pixel Space

Previously we examine the quantization error in latent space. We can further discuss the quantization error in pixel space given the decoder is smooth. More specifically, we have:

**Corollary 3.** *Following the setting in Theorem 3.1, and assuming the decoder $g(.)$ satisfy $|g(x_1) - g(x_2)| \leq c_3|x_1 - x_2|$, we have:*

$$\Pr\{|g(\hat{z}) - g(\mu)| \geq c_3\sigma_i\} \leq \exp(-e^t\sqrt{\frac{2}{\pi}}e^{-c_1-0.5}). \tag{34}$$

*Proof.* As $|g(\hat{z}_i) - g(\mu_i)| \leq c_3|\hat{z}_i - \mu_i|$, we have $\Pr\{|g(\hat{z}_i) - g(\mu_i)| \geq c_3\sigma_i\} \leq \Pr\{c_3|\hat{z}_i - \mu_i| \geq c_3\sigma_i\} = \Pr\{|\hat{z}_i - \mu_i| \geq \sigma_i\}$.

We can see that theoretically, the quantization error can be magnified by the Lipschitz constant $c_3$. However, we note that this is not a significant issue in practice. As shown in the Table 16, the actual loss of quality caused by GQ remains reasonable.

## D.3. Quantized Latent Visualization

In Figure 3, we show the t-NSE (van der Maaten & Hinton, 2008) visualization of latent after GQ, using 5 subclass of ImageNet.

*Table 15.* Quantitative results on COCO 2017 dataset. **Bold**: best.

| Method | bpp ($K \times N$) | UNet based | | | | ViT based | | | |
|--------|----------|-------|--------|-------|-------|-------|--------|-------|-------|
| | | PSNR↑ | LPIPS↓ | SSIM↑ | rFID↓ | PSNR↑ | LPIPS↓ | SSIM↑ | rFID↓ |
| VQGAN | | 26.25 | 0.099 | 0.756 | 14.110 | 25.11 | 0.106 | 0.747 | 11.231 |
| FSQ | | 26.01 | 0.072 | 0.767 | 5.451 | 25.85 | 0.112 | 0.765 | 11.213 |
| LFQ | 0.25 ($2^{16} \times 1024$) | 24.60 | 0.164 | 0.722 | 32.789 | 24.46 | 0.143 | 0.729 | 29.975 |
| BSQ | | 25.29 | 0.085 | 0.763 | 5.803 | 26.15 | 0.082 | 0.798 | 7.034 |
| GQ (Ours) | | **27.29** | **0.057** | **0.816** | **3.797** | **27.55** | **0.060** | **0.830** | **5.305** |
| VQGAN | | 29.06 | 0.049 | 0.839 | 6.616 | 27.83 | 0.058 | 0.832 | 5.461 |
| FSQ | | 29.08 | 0.043 | 0.855 | 4.008 | 28.51 | 0.053 | 0.851 | 5.390 |
| LFQ | 0.50 ($2^{16} \times 2048$) | 26.47 | 0.103 | 0.805 | 17.508 | 27.54 | 0.067 | 0.833 | 8.700 |
| BSQ | | 27.58 | 0.057 | 0.844 | 4.465 | 28.19 | 0.049 | 0.858 | 4.587 |
| GQ (Ours) | | **30.14** | **0.037** | **0.877** | **3.116** | **30.18** | **0.034** | **0.887** | **3.616** |
| VQGAN | | 31.97 | 0.024 | 0.901 | 3.455 | 31.07 | 0.029 | 0.904 | 3.494 |
| FSQ | | 32.30 | 0.022 | **0.917** | 2.797 | 31.48 | 0.023 | 0.911 | 3.045 |
| LFQ | 1.00 ($2^{16} \times 4096$) | 28.16 | 0.072 | 0.845 | 11.121 | 26.36 | 0.103 | 0.794 | 20.381 |
| BSQ | | 30.33 | 0.031 | 0.906 | 2.638 | 31.38 | 0.026 | **0.918** | 2.835 |
| GQ (Ours) | | **32.36** | **0.020** | 0.915 | **1.875** | **31.50** | **0.022** | 0.908 | **2.703** |

*Table 16.* The effect of quantization in pixel space.

| Latents | bits per latent | PSNR↑ | LPIPS↓ | SSIM↑ | rFID↓ |
|---------|-----------------|-------|--------|-------|-------|
| $\mu_i = \mathbb{E}[Z_i|X]$ (posterior mean) | 16 bits | 32.92 | 0.020 | 0.913 | 0.46 |
| $z_i \sim q(Z_i|X)$ (Gaussian sample) | $D_{KL}(q(Z_i|X)\|\mathcal{N}(0,1)) = 4.26$ bits | 32.61 | 0.021 | 0.911 | 0.46 |
| $\hat{z}_i$ (GQ) | $\log_2 K = 4$ bits | 32.11 | 0.023 | 0.906 | 0.414 |

## D.4. More Generation Results

To better understand the generation performance, in Table 17, we present the auto-regressive generation result of GQ in different bitrate regime. And in Table 18, we present the auto-regressive generation result of GQ in FFHQ dataset. It is shown that the advantage of GQ is consistent in different bitrate regime and datasets.

## D.5. Prior-Posterior Mismatch

Sometimes the Gaussian VAE might suffer from prior-posterior mismatch. However, in our case, such mismatch is not severe. To illustrate this, we estimate the prior posterior mismatch by considering the relationship between $q(Z)$ and $\mathcal{N}(0, I)$. More specifically, we have

$$D_{KL}(q(Z)\|\mathcal{N}(0,1)) \approx \frac{1}{N}\sum_{i=1}^{N}(\log q(z^i) - \log \mathcal{N}(z^i|0, I)). \quad (35)$$

*Table 17.* The generation performance of GQ in different bitrate regime.

| Method | bpp (num of tokens) | gFID | IS |
|--------|---------------------|------|-----|
| TokenBridge | 0.1875 ($2^{16} \times 256$) | 8.29 | 188.05 |
| GQ (Ours) | 0.1875 ($2^{16} \times 256$) | 7.74 | 229.53 |
| TokenBridge | 0.25 ($2^{16} \times 256$) | 7.82 | 198.24 |
| GQ (Ours) | 0.25 ($2^{16} \times 256$) | 7.67 | 230.79 |

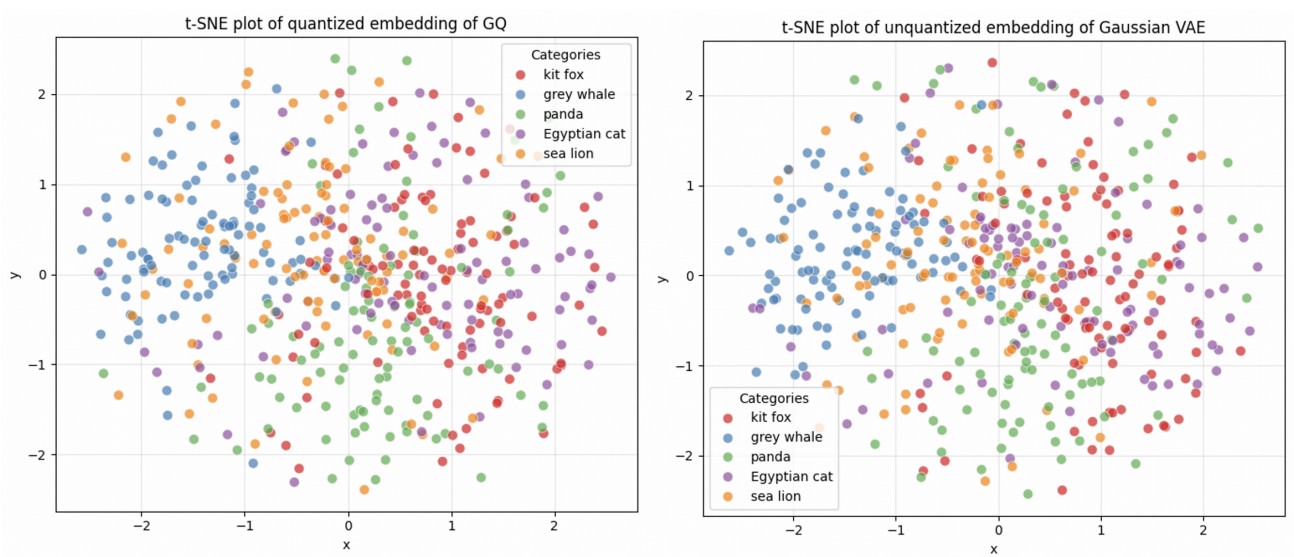

*Figure 3.* The t-NSE visualization of latent of GQ vs. unquantized Gaussian VAE. It is shown that the latent before and after quantization are quite similar.

*Table 18.* The generation performance of GQ with FFHQ dataset.

| Method | bpp (num of tokens) | gFID | IS |
|---|---|---|---|
| BSQ | $0.25(2^{16} \times 256)$ | 5.48 | - |
| TokenBridge | $0.25(2^{16} \times 256)$ | 7.15 | - |
| GQ (Ours) | $0.25(2^{16} \times 256)$ | 5.09 | - |

Additionally, we can estimate the optimal bitrate without effected by prior posterior mismatch with a similar approximation:

$$D_{KL}(q(Z|X)||p(Z)) \approx \frac{1}{N} \sum_{i=1}^{N} (\log q(z^i|X) - \log q(z^i)). \tag{36}$$

We train a diffusion model to estimate $\log q(z^i)$ by using PF-ODE and Skilling-Hutchinson trace estimator (See Appendix D.2 of Song et al. (2020)). We use the Gaussian VAE (w/ TDC) + DiT diffusion model and ImageNet validation dataset. The euler PF-ODE steps is set to 250 and the Skilling-Hutchinson number of sample is set to 1. The result is shown in Table 19. The results show that the prior-posterior mismatch is only 0.00033 bits-per-pixel, accounting for approximately 0.1% of the total bpp. Furthermore, the best bpp and the actual bpp show no significant difference on a scale of 0.01. This indicates that the "bitrate waste" caused by the prior-posterior mismatch is negligible, and the mismatch itself is not significant.

### D.6. Complexity

As with FSQ and BSQ (Mentzer et al., 2023; Zhao et al., 2024), our codebook can be generated on the fly by maintaining the same random number generator seed on both the encoder and decoder sides. Therefore, our GQ model has the same parameter size as the vanilla Gaussian VAE. In Table 20, we compare the encoding and decoding frames per second (FPS) of the Gaussian VAE and GQ. We use $256 \times 256$ images with a batch size of 1, and we report the wall clock time, meaning

*Table 19.* The bitrate and prior-posterior mismatch.

| Divergence | bits-per-pixel |
|---|---|
| bpp *w.r.t.* $\mathcal{N}(0, I)$ ($D_{KL}(q(Z||X)||\|\mathcal{N}(0,1))$) | 0.25 |
| bpp *w.r.t.* $q(Z)$ ($D_{KL}(q(Z||X)||\|q(Z))$) | 0.25 |
| prior posterior mismatch ($D_{KL}(q(Z)||\|\mathcal{N}(0,1))$) | 0.000328 |

*Table 20.* The encoding and decoding overhead of GQ over Gaussian VAE.

| Method | UNet based | |
| --- | --- | --- |
| | Encoding FPS | Decoding FPS |
| Gaussian VAE | 104 | 64 |
| GQ (torch) | 12 | 61 |
| GQ (CUDA) | 79 | 61 |

that the time required for loading data is included. The results show that the encoding FPS of GQ (implemented in PyTorch) is 12 on an H100 GPU, which is considerably slower than the 104 FPS achieved by the Gaussian VAE. On the other hand, GQ does not incur any decoding overhead.

To reduce the computational complexity of GQ, we implement GQ using a tailored CUDA kernel. Specifically, we follow the approach of Liu et al. (2025), with a key difference: we maintain the codebook, as our bottleneck is not codebook instantiation. Additionally, we avoid the creation of large buffer vectors by performing the summation over $m$ within the CUDA kernel instead of in PyTorch. With this approach, we achieve an encoding FPS of approximately 80, with negligible overhead compared to the Gaussian VAE. A detailed comparison between the PyTorch implementation and the CUDA implementation of GQ is provided below as `GQ_torch` and `GQ_CUDA`, respectively.

```
def GQ_torch(mu, sigma, codebook, m, bs, K):
    # mu.shape = (bs, m)
    # sigma.shape = (bs, m)
    # codebook.shape = (K, m)

    # This step create (bs, m, K) vector, which is the performance bottleneck
    dist_m =((mu[:,None] - codebook[None])/ sigma[:,None])**2
    dist = torch.sum(dist_m, dim=1) # sum over m dimension
    indices = torch.argmin(dist, dim=1) # argmax over K dimension
    zhat = torch.index_select(codebook, 0, indices) # select quantized results
    return indices, zhat

def GQ_CUDA(mu, sigma, codebook, m, bs, K):
    dist = torch.zeros([bs, K])
    # need an extension wrapping and register the kernel into operator, we omit it in paper
    # see code appendix for details
    GQ_Kernel<<<bs * K / 256,256>>>(mu, sigma, codebook, dist, m, bs, K)

    indices = torch.argmin(dist, dim=1) # argmax over K dimension
    zhat = torch.index_select(codebook, 0, indices) # select quantized results
    return indices, zhat

__global__ void GQ_Kernel(
  const float* mu,
  const float* sigma,
  const float* codebook,
  float* dist,
  int64_t m,
  int64_t bs,
  int64_t K
) {
  int idx = blockIdx.x * blockDim.x + threadIdx.x;
  if (idx >= K * bs) return;
  int bi = idx / K;
  int ni = idx % K;
```

```
    float a = 0.0f;
    for (int i = 0; i < m; i++) {
        float b = (codebook[ni * m + i] - mu[bi * dim + i]) / sigma[bi * dim + i];
        a += b * b;
    }
    dist[idx] = a;
    return;
}
```

### D.7. Asymptotic Complexity

It is noteworthy that GQ with codebook dimension $m = 1$, is asymptotically faster than reverse channel coding methods. This is because, for $m = 1$, the GQ target in Eq.3 reduces to a quadratic form. In this case, it suffices to sort the scalar codebook $c_{1:K}$ in advance. Despite the sorting takes $\Omega(D_{KL}(q(Z_i|X)||\mathcal{N}(0,1)))$, the sorting is only need to be done once and can be amortized across dimension and dataset. Subsequently, the minimization in Eq.3 can be performed in $O(D_{\mathrm{KL}}(q(Z_i|X)||\mathcal{N}(0,1)))$ time using binary search. The details is shown in Algorithm 2.

On the other hand, most reverse channel coding methods require $O(2^{D_{\mathrm{KL}}(q(Z_i|X)||\mathcal{N}(0,1))})$ computational complexity (Havasi et al., 2018b; Flamich et al., 2020; Theis & Yosri, 2021). A$^*$ coding (Flamich et al., 2022) can achieve $O(D_\infty(q(Z_i|X)||\mathcal{N}(0,1)))$ encoding complexity, albeit at the cost of increased decoding complexity.

However, we note that this complexity advantage is not particularly meaningful in practice. This is because any auto-regressive generation model requires a softmax operation over the entire codebook, which has a complexity of $O(2^{D_{\mathrm{KL}}(q(Z_i|X)||\mathcal{N}(0,1))})$. In practice, only tractable codebook sizes, such as $2^{16}$ or $2^{18}$, are used.

## E. Additional Quantitative Results

### E.1. Additional Qualitative Results and Failure Cases

In Figure 4, we present additional qualitative results showing that GQ achieves superior visual quality. However, we also note that none of the approaches is successful in reconstructing the license of the residential vehicle. The text content remains challenging for low bitrate VQ-VAEs.

---

**Algorithm 1** GQ (Argmax)

**Require:** Codebook $c_{1:K}$ (sorted, $c_j \leq c_{j+1}$), $\mu_i$
**Ensure:** Quantized value $\hat{z}_i$, index $j^*$
1: $T^* \leftarrow \infty$
2: **for** $j = 1$ to $K$ **do**
3:    **if** $T^* \leq \|c_j - \mu_i\|$ **then**
4:       $T^* \leftarrow \|c_j - \mu_i\|$
5:       $j^* \leftarrow j$
6:    **end if**
7: **end for**
8: **return** $c_{j^*}, j^*$

---

**Algorithm 2** GQ (Bisect)

**Require:** Codebook $c_{1:K}$ (sorted, $c_j \leq c_{j+1}$), $\mu_i$
**Ensure:** Quantized value $\hat{z}_i$, index $l$ or $r$
1: $l \leftarrow 1, r \leftarrow K$
2: **while** $l + 1 < r$ **do**
3:    $m \leftarrow \lfloor (l+r)/2 \rfloor$
4:    **if** $c_m < \mu_i$ **then**
5:       $l \leftarrow m$
6:    **else**
7:       $r \leftarrow m$
8:    **end if**
9: **end while**
10: **if** $\|c_l - \mu_i\| < \|c_r - \mu_i\|$ **then**
11:    **return** $c_l, l$
12: **else**
13:    **return** $c_r, r$
14: **end if**

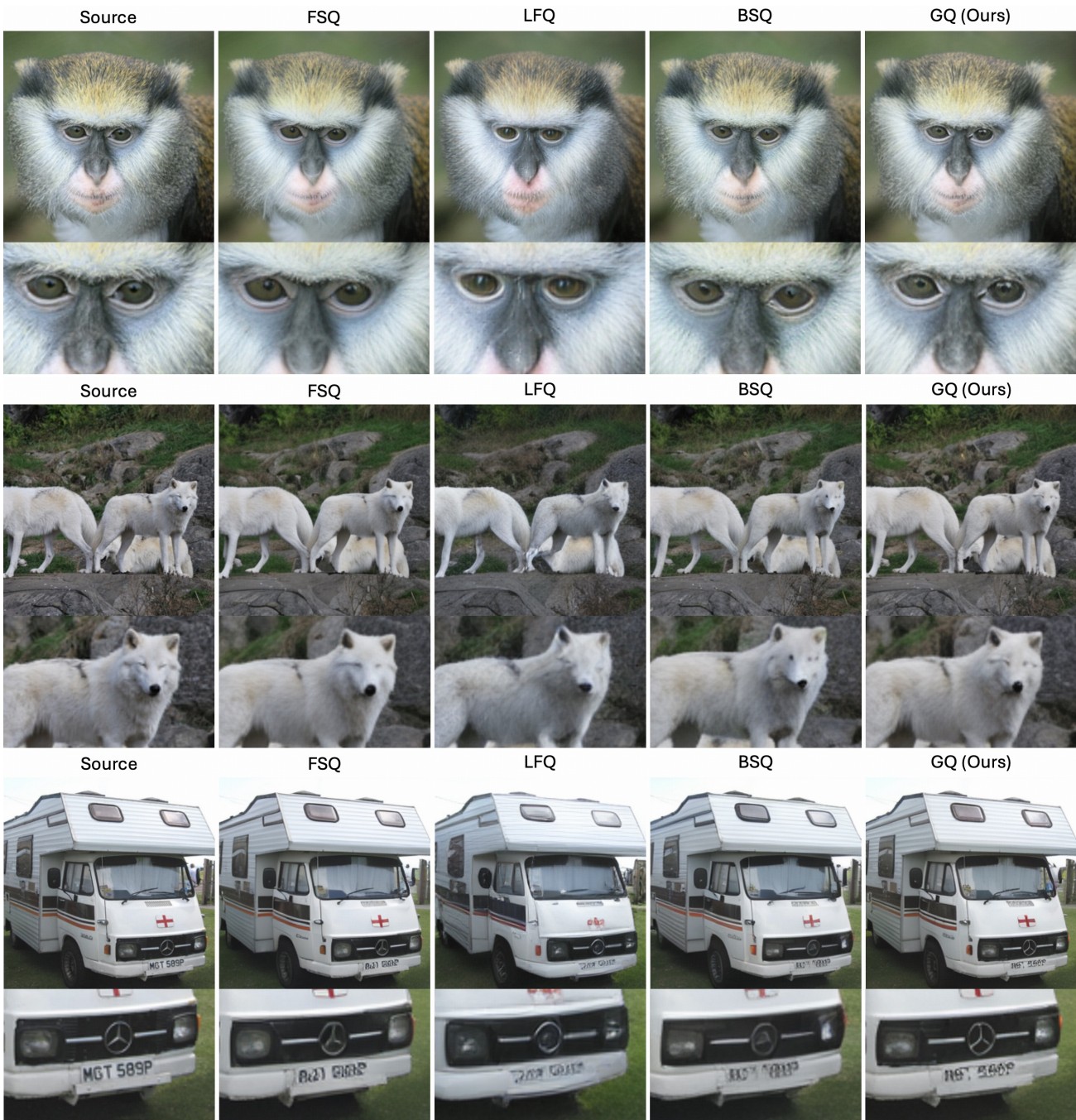

*Figure 4.* Qualitative results on ImageNet dataset and 0.25 bpp. None of these approaches correctly reconstructs the license plate.

