# OpenReview forum: "Training-Free Vector Quantization via Gaussian VAEs"
_ICML.cc/2026/Conference — ICML 2026 regular_

### Official Review · Reviewer_dBoG · 2026-02-22

**Soundness:** 3
**Presentation:** 2
**Significance:** 3
**Originality:** 2
**Overall Recommendation:** 4
**Confidence:** 3

**Summary:**

This paper proposes Gaussian Quant, a method that converts a conventional Gaussian VAE into a VQ-VAE without requiring additional training. While existing VQ-VAE models are often difficult to train, the proposed approach leverages a Gaussian VAE and transforms it into a discretized latent representation by leveraging the target divergence constraint, which regularize the KL-divergence term. A key contribution of the paper is the theoretical analysis of the properties required for $K$, one of the central parameters in discretized latent space models. The authors also demonstrate empirically that the proposed approach outperforms existing discrete latent space models.

**Compliance With Llm Reviewing Policy:**

Affirmed.

**Key Questions For Authors:**

1. At the beginning of Section 3.2, the paper states that the choice of $K$ is not obvious and then presents Theorems 3.1 and 3.2. However, as I understand it, these theorems provide theoretical results regarding the range of plausible values for $K$, rather than a principle for selecting a specific value. In practice, the experiments still perform hyperparameter search over values from $2^{14}$ to $2^{18}$. It would therefore be helpful if the authors could clarify how these theoretical results translate into guidance for choosing a specific $K$ in practice.

2. In the final part of Section 3.2, the paper states that having a small $K$ is more problematic from the perspective of reconstruction error, while a sufficiently large $K$ makes the codebook very dense in the continuous space. As $K$ increases, the value of $R_i$ would also increase, which in turn would push the posterior variance $\sigma^2$ closer to 1 (i.e., increase it). In that case, would $|\hat{z} - \mu|$ almost always remain smaller than $\sigma$, implying that large quantization errors rarely occur in the first place?

3. While the paper proposes a meaningful approach that makes discrete latent space models such as VQ-VAE easier to train, Table 1 suggests that the continuous Gaussian VAE (before applying the Gaussian Quant) already achieves outperforming performance. Given this, it would be helpful if the authors could elaborate on the motivation for discretizing the latent space in this setting.

4. As a follow-up to Question 3, if the continuous Gaussian VAE already achieves stronger performance, it is unclear whether using a very large $K$ (so that the codebook becomes extremely dense in the continuous latent space) can still be meaningfully interpreted as a discretization of the latent space. In that case, if much of the performance gain comes from the well-trained Gaussian VAE rather than from the GQ step itself, it might suggest that the primary contribution of the paper lies more in TDC than in GQ. I would be interested in the authors' perspective on this point, particularly given that Appendix Table 13 shows that GQ uses the largest codebook size (2^{16}).

**Limitations:**

yes

**Strengths And Weaknesses:**

### Strengths
1. The experiments are organized in a very systematic manner, including comprehensive ablation studies.

2. The paper theoretically characterizes the properties required of $K$, an important hyperparameter in models with discrete latent spaces.

3. The work proposes a practical methodology for deriving VQ-VAE style models (which have traditionally been difficult to train) by leveraging the more easily trainable Gaussian VAE model.

### Weaknesses
1. Several aspects of the paper would benefit from clarification; please see "Key Questions for Authors" section.

---

> ### Author Rebuttal · Authors · 2026-03-29
>
> Thank you for your feedback. We are glad you find our work is well organized with comprehensive empirical study, characterizes theoretical prperty for discrete VAE and solves a practical problem. Below we address the concerns in detail.
>
> ### Q1. How these theoretical results translate into guidance for choosing a specific K
> * We run the GQ ImageNet experiment at 0.25 bpp ($2^{16} \times 1024$) and visualize the bounds from __Theorem 3.1__ and __Theorem 3.2__ at https://ibb.co/ym4DD9xm. These bounds indicate that we should select $D_{KL} \le \log K \le D_{KL} + 5$. Empirically, we find that the lowerbound, i.e., $\log K = D_{KL}$ works very well (see Table 7). We have not searched $K$ over $2^{14}$ to $2^{18}$; we simply set $\log K=D_{KL}$.
> * The importance of __Theorem 3.1__ and __Theorem 3.2__ lies in bounding the value of $K$ to $D_{KL}$, rather than determining a single universal value of $\log K$ that is suitable for all conditions. In practice, $K$ is determined first by user, as it is an input parameter of VQ-VAE. The Gaussian VAE is then trained with a target $D_{KL} = \log K$. Without __Theorem 3.1__ and __Theorem 3.2__, we have no understanding of the relationship between $D_{KL}$ and $K$. With a vanilla Gaussian VAE, $D_{KL}$ varies greatly per dimension, and no matter how much parameter search is performed, the resulting GQ would fail (see Table 5).
>
> ### Q2. When $R_i$ is large, would $|z - \mu|$ almost always smaller than $\sigma$, implying that large quantization errors rarely occur?
> * First off, the statement __"a large $R_i$ pushes $\sigma_i$ towards 1"__ is incorrect. In fact, when $\mu_i$ is fixed, a small $R_i$ pushes $\sigma_i$ towards $1$. This is because $R_i = D_{KL}$ is defined as
>     $$
>     R_i = \frac{1}{2}(\mu_i^2+\sigma_i^2-\log \sigma_i^2 - 1).
>     $$
> * For fixed $\mu_i$, $R_i$ is minimized by $\sigma_i = 1$, not maximized by $\sigma_i = 1$. In the scenario of a Gaussian VAE, a large $R_i$ pushes $\sigma_i$ toward $0$ [Fast Relative Entropy Coding with A* Coding] [Minimal Random Code Learning with Mean-KL Parameterization]. Therefore, the intuition from the above statement, __"large quantization errors rarely occur for large $K$"__, is also incorrect. In fact, a large $R$ requires a even smaller quantization error to achieve $|\hat{z} - \mu| \le \sigma$.
> * Empirically, we show a case where $\log K$ is sufficiently large (16), while $R_i$ is larger (24). It is shown that the probability of large quantization error is close to 100%, and the quantization error becomes quite significant (PSNR from 29.66 to 27.46 dB).
>
>     | | bpp | Large quantization error | PSNR | LPIPS | SSIM | rFID |
>     |-|-|-|-|-|-|-|
>     | Gaussian VAE w/ TDC | 0.375 ($2^{24}\times 1024$)  | - | 29.66 | 0.046 | 0.854 | 0.517 |
>     | GQ | 0.25 ($2^{16}\times 1024$) | 97% | 27.46 | 0.074 | 0.805 | 0.642 |
>
> ### Q3. Continuous Gaussian VAE already achieves good performance
> * Continuous Gaussian VAE achieves the best reconstruction. However, this does not eliminate the necessity of discrete VAEs.
>   * In terms of image compression, the bpp (bits per dimension) of a Gaussian VAE is not directly achievable, and the bits-back coding bitrate remains a theoretical value when the VAE is lossy. In contrast, the bpp of a discrete VAE is achievable, and images can be directly stored into files with the theoretical bpp.
>   * In terms of image generation, a continuous Gaussian VAE cannot be used for autoregressive generation, whereas a discrete VAE can. As shown in Table 6, discrete VAE with autoregressive generation is more efficient than Gaussian VAE with diffusion generation, a finding also verified in [Autoregressive Model Beats Diffusion: Llama for Scalable Image Generation].
>
> ### Q4. using a very large codebook ($2^{16}$) can still be interpreted as a discretization ?
> * When the codebook size is $K = 2^{16}$, we use a codebook dimension of $16$ for 0.25 bpp. This means we quantize $R^{16}$ into $2^{16}$ pieces, rather than $R^1$. This is a fairly sparse and discrete representation: if we combine 16 binary quantized 1D latents, we obtain a 16-dimensional latent with a codebook size of $2^{16}$. This is clearly a meaningful discretization.
> * The codebook sizes $K = 2^{16}$ and dimension $16$ are chosen to align with standard VQGAN (see Table 13). FSQ, LFQ and BSQ has small codebook size (LFQ: $2^1$, FSQ: $2^4$, BSQ: $2^1$) and lower codebook dimension (LFQ: 1, FSQ: 1, BSQ: 1). For generation, their small tokens are also grouped into one large token with codebook size $2^{16}$ and higher dimension to match VQGAN. For the final $2^{16}$ large token, VQGAN, LFQ, BSQ and GQ have same codebook dimension $16$, only FSQ has codebook dimension $4$, which is more "discrete" than others.
>
> We hope these improvements address your concerns. Thank you again for your careful reviews.

---

> > ### Author Rebuttal · Reviewer_dBoG · 2026-04-01
> >
> > I appreciate the authors' efforts in addressing my concerns. Many aspects have become much clearer following the rebuttal.
> >
> > I will therefore increase my score from 3 to 4.

---

> > > ### Author Response · Authors · 2026-04-08
> > >
> > > We greatly thank the reviewer again for their support of our paper. We're glad to be able to address your questions.

---

### Official Review · Reviewer_iE8t · 2026-02-27

**Soundness:** 3
**Presentation:** 4
**Significance:** 3
**Originality:** 3
**Overall Recommendation:** 4
**Confidence:** 4

**Summary:**

This submission centers on a critical technical proposition: circumventing the well-documented, notorious training difficulties of VQ-VAEs via a two-stage pipeline that combines constrained Gaussian VAE training and training-free conversion to discrete tokenizers. To address the long-standing core pain points of VQ-VAEs, the authors propose the GQ method. This approach completely circumvents the optimization pitfalls of end-to-end VQ-VAE training through a two-stage paradigm: Gaussian VAE pre-training with TDC, paired with training-free quantization using a fixed Gaussian codebook. The manuscript provides rigorous theoretical proofs for codebook size selection and quantization error bounds, and conducts comprehensive experimental validation across two mainstream architectures, UNet and ViT. Results show that GQ comprehensively outperforms existing SOTA methods including VQGAN, FSQ, LFQ, and BSQ on both image reconstruction and generation tasks, while also delivering performance improvements for peer methods in the same field such as TokenBridge. Overall, the core theme explored in this work is to establish a vector quantization paradigm with rigorous theoretical guarantees and straightforward practical usability, breaking the long-standing trade-off between training stability and rate-distortion performance in the VQ-VAE field.

**Compliance With Llm Reviewing Policy:**

Affirmed.

**Key Questions For Authors:**

Could you explain why this staged optimization approach outperforms the joint optimization scheme used in VQ-VAE? Is it purely engineering challenges that make VQ-VAE so difficult to train effectively? And what are the theoretical upper performance bounds for these two methods?

**Limitations:**

Yes

**Strengths And Weaknesses:**

## Strengths
1.  **Solid theoretical contribution that fills a critical gap in existing work**
    This work provides a rigorous mathematical proof for the relationship between quantization error and codebook size: the probability of large quantization errors decays double-exponentially when the codebook rate exceeds the bit-back coding rate of Gaussian VAE.

2.  **Clean, elegant method design that addresses the core pain points of VQ-VAE at their root**
    The proposed two-stage paradigm fully decouples two tightly coupled, conflicting tasks: optimal continuous-domain representation learning and discrete quantization. The first-stage Gaussian VAE training is a well-established continuous optimization problem, with no gradient approximation bias and rigorous convergence guarantees. The training-free quantization in the second stage completely eliminates the issues of STE gradient distortion and the deadlock of joint codebook updates. The method requires no complex regularizer design, features a simple and adaptive TDC constraint, and has an extremely low barrier to engineering implementation.

3.  **Rigorous, comprehensive experimental design with highly credible conclusions**

4.  **Outstanding practical value with exceptional deployment and generalization capabilities**
    The work demonstrates that GQ not only achieves SOTA performance on its own, but its core TDC constraint can also directly boost the performance of existing Gaussian VAE discretization methods such as TokenBridge, showing extremely strong generalization. Meanwhile, GQ has high robustness to training hyperparameters, no risk of codebook collapse, and far lower computational overhead than end-to-end VQ-VAE. Its codebook can be generated in real time with no additional storage cost, and can be directly and seamlessly embedded into the visual Tokenizer frameworks of existing autoregressive generative models and diffusion models, delivering extremely high value for industrial deployment.

## Weaknesses
1.  **Insufficient analysis of boundary cases in the theoretical section**
    For Theorems 3.1 and 3.2, only the upper bound expressions for c1 and c2 are provided, with no numerical ranges under typical experimental settings, nor an analysis of the tightness of the theoretical upper bounds in extremely low code rate (<0.1 bpp) and extremely high code rate (>1.0 bpp) scenarios. Meanwhile, the multi-dimensional codebook extension in Section 3.4 only presents the engineering implementation, without supplementing the theoretical bound of quantization error for this scenario, leaving room for further improvement in theoretical completeness.

2.  **Limited depth of exploration for generative tasks**
    The paper focuses heavily on image reconstruction performance, yet the core deployment scenarios for current visual Tokenizers are large-scale generative tasks. The work only includes simple class-conditional image generation experiments, and does not validate GQ’s performance in complex tasks such as text-to-image, high-resolution generation, and video generation. It also does not provide an in-depth analysis of the gap between reconstruction performance and generation performance. While the paper notes this will be addressed in future work, adding 1-2 sets of relevant validation experiments would significantly strengthen the persuasiveness of the method for a top-tier conference submission.

3.  **Lack of in-depth comparative analysis in the related work section**
    For fixed-codebook VQ-VAE methods such as FSQ and BSQ, the paper only presents experimental performance comparisons, without an in-depth analysis of the fundamental differences between GQ and these methods — including core distinctions in optimization objectives, theoretical foundations, and rate-distortion characteristics. For work related to backward channel coding, the authors only compare computational complexity, and do not conduct more in-depth comparative analysis across dimensions such as quantization error and adaptability to deterministic quantization. The discussion of related work can be further deepened.

4. The training process for the Gaussian VAE has inherent built-in constraints — this means the phrasing "given a Gaussian VAE" in Line 135 may not be the most appropriate.

---

> ### Author Rebuttal · Authors · 2026-03-29
>
> Thank you for your feedback. We are glad you find our work fills a critical theoretical gap, with elegant method and outstanding practical value. Below we address the concerns in detail.
>
> ### W1. Insufficient theoretical analysis
> * We run the GQ ImageNet experiment at 0.25 bpp ($2^{16} \times 1024$), compute the values $c_1 = 5.26, c_2 = 1.50$, and visualize the bounds from __Theorem 3.1-3.2__ at https://ibb.co/ym4DD9xm. Those bounds indicates that we should select $D_{KL} \le \log K \le D_{KL} + 5$. Empirically, we find that $\log K = D_{KL}$ works very well (see Table 7).
> * For more extreme bitrates such as 1.00 bpp ($2^{16} \times 4096$), the values are $c_1 = 5.14$ and $c_2 = 1.44$, showing no significant change. This is because the bitrate is mainly controlled by the number of tokens, which does not affect $c_1,c_2$.
>
> ### W2. The multi-dimensional bound
> * We can extend __Theorem 3.1-3.2__ to multi-dimensional and get similar conclusion.  __Due to the 5000 character limit, we only provide proof sketch__.
> * __Corollary 3.1__ Let $\mu,\sigma \in R^d$ be the mean and std of d dimensional factorized Gaussian $q(Z|X)$. Assume $||\mu\sigma||\le c_1,||\mu||+||\sigma||\le c_2,\prod_{k=1}^d \sigma_k \ge c_3$. Denote $R = D_{KL}(q(Z|X)||N(0,I_d))$, $c_1,...,c_K\sim N(0,I_d)$, quantization error $\mathcal{E} = \min_{j=1,...,K} ||\frac{\mu_i - c_j}{\sigma_i}||$. Then we have
>     $$
>     \log K = R + t, \\
>     P(\mathcal{E} \ge \sqrt{d}) \le \exp{(-e^{t}C(d,c_1,c_2,c_3))},
>     $$
>     where $C(d,c_1,c_2,c_3)$ is some constant determined by $d,c_1,c_2,c_3$.
> * __proof sketch__:
>     * We first rewrite the event:
>         $$
>         P(\mathcal{E} \ge \sqrt{d}) \le \exp{(-K P(S))}, \textrm{where } S = \{x\in R^d:||\frac{x - \mu}{\sigma}|| \le \sqrt{d}\}.
>         $$
>     * Similar to __Theorem 3.1__, we can bound $P(S)$ by the Mean value theorem for integrals:
>         $$
>         P(S) = \int_{S} \Phi_d(x) dx = \Phi_d(x')V(S).
>         $$
>     * We first bound the volume:
>         $$
>         V(S) \ge V(B_d(1)) \sqrt{d}c_3.
>         $$
>     * Next, we bound the density:
>         $$
>         \Phi_d(x') \ge \frac{1}{(2\pi)^{d/2}}\exp{(-R_i)}\exp{(-\frac{d - d\log c_2^2 - (d-1)c_2^2 - 2\sqrt{d}c_1}{2})}.
>         $$
>     * And the final bound becomes:
>         $$
>         P(\mathcal{E} \ge \sqrt{d}) \le \exp{(-e^{t}\underbrace{\frac{1}{(2\pi)^{d/2}}\exp{(-\frac{d - d\log c_2^2 - (d-1)c_2^2 - 2\sqrt{d}c_1}{2})}  V(B_d(1)) \sqrt{d}c_3}_{C(d,c_1,c_2,c_3)})}.
>         $$
> * __Corollary 3.2__ Following the setting in __Corollary 3.1__, we have
>     $$
>     \log K = R - t, \\
>     P(\mathcal{E} \ge \sqrt{d}) \ge 1 -e^{-t}C(d,c_1,c_2,c_3).
>     $$
> * __proof sketch__:
>     * Following __Theorem 3.2__ and plug in $P(S)$ in __Corollary 3.1__
>         $$
>         P(\mathcal{E} \ge \sqrt{d}) = (1 - P(S))^K \ge 1 - K P(S) \ge 1 -e^{-t}C(d,c_1,c_2,c_3).
>         $$
>
> ### W3. Limited depth of exploration for generative.
> * For large scale generation model, see Q1 of aZZv.
> * Both autoregressive and diffusion generation paper suggest that reconstruction is not the most important factor for generation [Scaling Visual Tokenizers to 3 Billion Parameters for Autoregressive Image Generation][VAVAE: Taming Optimization Dilemma in Latent Diffusion Models]. The real factor is the smoothness of the latent space, which can be optimized by aligning to DINO. Below, we provide experimental results for GQ with the DINO alignment loss proposed in [VAVAE]. It is shown that DINO alignment significantly affects generation performance, even when reconstruction performance is similar.
>
>     | | PSNR | DINO Align | gFID |
>     |-|-|-|-|
>     | Gaussian VAE w/ TDC + Diffusion | 27.83 | No | 8.47 |
>     | Gaussian VAE w/ TDC  + Diffusion | 27.50 | Yes | 5.31 |
>     | GQ + AR | 27.26 | No | 7.67 |
>     | GQ + AR | 26.73 | Yes | 4.19 |
>
> ### W4. in-depth comparative analysis in the related work section
> * In revised paper, we provide an additional analysis covering related methods. __Due to the 5000 character limit, here we only provide a summary table of optimization techniques for discrete VAEs.__
>     | | STE | Entropy Loss | Codebook Loss | Commitment Loss |
>     |-|-|-|-|-|
>     | VQGAN | Yes | No | Yes | Yes |
>     | FSQ | Yes | No | No | No |
>     | BSQ | Yes | Yes | No | No |
>     | LFQ | Yes | Yes | No | Yes |
>     | GQ | No | No | No | No |
> * For reverse channel coding, we provide quantization error in Table 4. As discussed in the related works, reverse channel coding are not applicable to deterministic quantization.
>
> ### W5. The phrasing "given a Gaussian VAE"
> * In revised paper, we amend it into: "given a Gaussian constraint VAE".
>
> ### Q1. staged vs joint optimization?
> * Stage-optimized GQ outperforms jointly optimized VQ-VAE, as GQ eliminates the need for gradient approximation caused by discrete training. However, GQ can also be jointly optimized and achieves a performance gain (See Table 9).
>
> We hope these improvements address your concerns. Thank you again for your careful reviews.

---

> > ### Author Rebuttal · Reviewer_iE8t · 2026-04-02
> >
> > Thanks for the detailed rebuttal. I appreciate the effort in providing the $c_1$ and $c_2$ values, the multi-dimensional proof sketches (Corollaries 3.1 and 3.2), and addressing the related work comparisons. The extra DINO experiments were also helpful and cleared up my concerns. While the response effectively answers my questions and strengthens the paper's theoretical completeness, my overall view on its impact remains the same. It's a technically solid paper with a clean method and strong theoretical foundation. However, to push the contribution higher, it still needs a more comprehensive exploration of large-scale, complex generative tasks (like text-to-image or high-res video generation). Because of this, I'm keeping my score as a Weak Accept.

---

> > > ### Author Response · Authors · 2026-04-08
> > >
> > > We greatly thank the reviewer again for their support of our paper. We're glad to be able to address your questions. Despite scaling up autoregressive generation to text-to-image or high-res video is beyond our current resource constraints, we will address this in future works.

---

### Official Review · Reviewer_KZXN · 2026-03-05

**Soundness:** 2
**Presentation:** 1
**Significance:** 3
**Originality:** 3
**Overall Recommendation:** 5
**Confidence:** 2

**Summary:**

This paper proposes a simple alternative to training VQ-VAEs. Instead of directly optimizing a discrete tokenizer, the authors first train a continuous Gaussian VAE under a rate-distortion objective, with a per-dimension KL constraint (Target Divergence Constraint, TDC) to control the information allocation. After training, they convert the continuous latent space into a discrete one by quantizing each dimension using a fixed Gaussian codebook whose size matches the KL rate. The resulting discrete autoencoder matches or exceeds strong VQ-based baselines across multiple backbones and bitrates, without requiring complex VQ training procedures.

**Compliance With Llm Reviewing Policy:**

Affirmed.

**Final Justification:**

Based on the answers provided and the additional results, I confirm my previous opinion on the significance and importance of the paper's contribution. However, improvement in writing and presentation is strongly advised for the revised version.

**Key Questions For Authors:**

- The paper focuses primarily on the advantages of the proposed approach. Could the authors discuss the main practical downsides or limitations of the method? For example, are there stability issues during training, increased computational cost compared to standard VQ training, or sensitivity to hyperparameters introduced by TDC?
- The theoretical results introduce constants c1 and c2 governing the decay of quantization error as a function of log K-Ri. Could the authors provide more intuition on how these constants behave in practice? For instance, how tight are these bounds empirically, and how much slack is needed between log K and the per-dimension KL to ensure reliable discretization?
- How does the choice of latent dimensionality (number of latent tokens) interact with TDC? Does increasing the number of latent dimensions change the optimal KL target or the stability of discretization?

**Limitations:**

No, see questions

**Strengths And Weaknesses:**

- Soundness: The core idea of training a Gaussian VAE under a rate-distortion objective and converting it post hoc into a discrete model has a clear information-theoretic motivation. The Target Divergence Constraint (TDC) is well aligned with the theoretical argument that successful quantization requires matching the per-dimension KL to the codebook size. The theoretical claims are reasonable and consistent with rate-distortion intuition, and the empirical section supports them with controlled ablations.
Experimentally, the method performs competitively across multiple backbones and bitrates, and the gains over strong VQ-style baselines are consistent. The authors also evaluate generative performance when the discrete tokens are used for downstream modeling, which strengthens the practical relevance of the tokenizer.
A potential weakness on the technical side is the number of hyperparameters introduced by TDC (target KL, tolerance band, adaptive weights), which may affect robustness in practice.

- Presentation: The main weakness of the paper is presentation. The writing is often difficult to follow, with dense explanations and abrupt transitions. Some conceptual steps such as the intuition behind TDC and its practical update rule could be explained more cleanly and earlier in the paper. As written, the contribution feels more complicated than it actually is. Structurally, the paper would benefit from a clearer high-level roadmap and a more explicit comparison to related approaches.

- Significance: The paper addresses the relevant and practical problem of training discrete visual tokenizers, which is known to be unstable and cumbersome. Proposing a simpler alternative that allows to train a continuous VAE and discretize afterward is conceptually appealing and practically useful. If robust, this approach could simplify the design of visual tokenizers used in autoregressive and multimodal generative models. The demonstrated generative performance further strengthens its impact.

- Originality: The proposed approach is novel and clearly distinguishable from prior VQ-based training pipelines. Instead of directly learning a discrete codebook, the method introduces a principled way to control per-dimension information in a Gaussian VAE and then performs post-training quantization. The Target Divergence Constraint provides a concrete mechanism that enables this conversion to work reliably in practice.

---

> ### Author Rebuttal · Authors · 2026-03-29
>
> Thank you for your feedback. We are glad you find our work as conceptually appealing and practically useful, with reasonable theoretical claims and competitive empirical results. Below we address the concerns in detail.
>
> ### W1. The intuition behind TDC and its practical update rule could be explained more cleanly and earlier in the paper, the paper would benefit from a clearer high-level roadmap and a more explicit comparison to related approaches.
> * As discussed in the related works section, prior work such as [MIRACLE: Compressing Images by Encoding Their Latent Representations with Relative Entropy Coding][High-Fidelity Generative Image Compression] have already demonstrated how to constrain the mean of $R_i$ given a target bitrate. The idea is to adjust $\lambda_i$ based on the mean of $R_i$ and the target $R_{target}$:
>     $$
>     \lambda_i = \lambda_i * \alpha \textrm{ if mean}(R_i) \ge R_{target},
>     $$
>     $$
>     \lambda_i = \lambda_i / \alpha \textrm{ if mean}(R_i) \le R_{target},
>     $$
> * TDC is an extension of such approaches, effectively constraining the minimum and maximum of $R_i$ in addition to the mean.
> * In revised paper, we move this discussion from the related work section to the beginning of Sec 3.3.
>
> ### Q1. Could the authors discuss the main practical downsides or limitations of the method ?
> * In terms of training stability, our GQ is more stable than VQ and does not suffer from codebook collapse. Moreover, GQ is faster to train than VQ and achieves better convergence under the same computational budget (see Table 1 and Table 2). TDC is also quite robust to hyperparameters (see Table 10). Compared with VQ, the main disadvantage of GQ is that TDC indeed introduces several additional hyperparameters, making GQ harder to implement and understand. However, we hope to mitigate this limitation through our open-sourced code and paper.
>
> ### Q2. Could the authors provide more intuition on how these constants behave in practice? For instance, how tight are these bounds empirically, and how much slack is needed between log K and the per-dimension KL to ensure reliable discretization?
> * We run the GQ ImageNet experiment at 0.25 bpp ($2^{16} \times 1024$), compute the practical values of $c_1 = 5.26$ and $c_2 = 1.50$, and visualize the bounds from __Theorem 3.1__ and __Theorem 3.2__ at https://ibb.co/ym4DD9xm. It is shown that the upper bound in Theorem 3.1 is tight, and the probability of large quantization error diminishes when $\log K \ge D_{KL} + 5$. On the other hand, the lower bound in Theorem 3.2 is tighter, and when $\log K$ is smaller than $D_{KL}$, the probability of large quantization error increases rapidly. Those bounds indicates that we should select $D_{KL} \le \log K \le D_{KL} + 5$. Empirically, we find that $\log K = D_{KL}$ works very well (see Table 7).
>
> ### Q3. Does increasing the number of latent dimensions (number of latent tokens) change the optimal KL target or the stability of discretization?
> * Based on our experiments with the number of latent tokens in the range [256, 4096], TDC and GQ remain quite stable within that range. We further tested GQ with an even larger number of tokens, such as 8192, at a target bitrate of 16. The results show that TDC control remains effective and GQ discretization remains robust.
>
>     | | bpp | R (mean, min - max) | PSNR | LPIPS | SSIM | rFID |
>     |-----------------|---------------------------------------|-----------|------|-------|------|------|
>     | Gaussian VAE w/ TDC    | 2.00  |  15.9, 14.7 - 17.1  | 35.88 | 0.010 | 0.950 | 0.21 |
>     | GQ    | 2.00 ($2^{16}\times 8192$) | - | 35.78 | 0.010 | 0.948 | 0.28 |
>
> We hope these improvements address your concerns. Thank you again for your careful reviews.

---

> > ### Author Rebuttal · Reviewer_KZXN · 2026-04-02
> >
> > I thank the authors for answering my questions and addressing my concerns. I find this method very relevant and promising, but the paper could benefit from improved writing and presentation. I raised my score to 5, but strongly encourage the authors to improve the writing and add a clear limitation section in the revised version.

---

> > > ### Author Response · Authors · 2026-04-08
> > >
> > > We greatly thank the reviewer again for their support of our paper. We're glad to be able to address your questions. We will revise our paper accordingly.

---

### Official Review · Reviewer_aZZv · 2026-03-11

**Soundness:** 4
**Presentation:** 3
**Significance:** 3
**Originality:** 3
**Overall Recommendation:** 5
**Confidence:** 4

**Summary:**

This paper proposes Gaussian Quant (GQ), a simple and effective framework that converts a constrained Gaussian VAE into a VQ-VAE without additional codebook training. The key idea is to quantize posterior means using a Gaussian codebook, together with a target divergence constraint (TDC) that aligns per-dimension KL rates with the target codebook bitrate. The method is well motivated theoretically and delivers strong reconstruction results across both UNet and ViT backbones, while also improving prior Gaussian-VAE-based discretization methods such as TokenBridge.

**Compliance With Llm Reviewing Policy:**

Affirmed.

**Final Justification:**

After reading the full set of reviewer comments and the authors’ rebuttal, I believe my main concerns have been sufficiently addressed. I appreciate the authors’ effort in clarifying the paper and responding thoughtfully to the feedback. As a result, I have decided to raise my score from 4 to 5.

**Key Questions For Authors:**

The current evidence is strongest for reconstruction quality. Do the authors expect the advantage of GQ to persist when scaling up downstream autoregressive generation, especially under stronger generator training budgets?

Since TDC is central to making shared codebooks work well, how robust is the method when the target bitrate or codebook size is misspecified relative to the intrinsic per-dimension rates of the Gaussian VAE?

Have the authors considered integrating GQ into multi-scale or residual tokenization frameworks, and if so, do they expect the same rate-matching intuition to transfer cleanly to those settings?

Could the authors also report precision and recall in Table 6, following the evaluation protocol used in TokenBridge Table 3?

If the authors can satisfactorily address the above concerns, I would be willing to raise my score.

**Limitations:**

yes

**Strengths And Weaknesses:**

The paper is conceptually clean and addresses an important practical issue: the difficulty of training VQ-VAEs directly. I especially like that the method is not just an empirical trick; the analysis relating quantization quality to the bits-back coding rate gives a clear principle for choosing the codebook size. The experiments are also convincing on reconstruction, showing consistent gains over strong baselines on multiple architectures, and the fact that TDC also improves TokenBridge makes the contribution feel broader than a single method.

My main reservation is that the paper is much stronger on reconstruction than on generation. The authors also acknowledge that strong reconstruction does not necessarily imply strong generation, and the evaluation does not fully establish whether GQ is the best tokenizer for downstream generative modeling. In addition, the comparisons focus on a standard single-scale setting, so it remains somewhat unclear how the method would fare against the strongest multi-scale or residual tokenizers under fully matched conditions.

While the generation results are promising, the evaluation remains somewhat incomplete because it reports only gFID and IS, without precision/recall. A more comprehensive generative evaluation would strengthen the paper’s evidence that GQ is competitive as a tokenizer for downstream generative modeling.

---

> ### Author Rebuttal · Authors · 2026-03-29
>
> Thank you for your feedback. We are glad you find our work conceptually clean and solid with analysis plus convincing experiments. Below we address the concerns in detail.
>
> ### Q1. Do the authors expect the advantage of GQ to persist when scaling up downstream autoregressive generation ?
> * Although we cannot currently scale up autoregressive generation due to resource constraints, we are confident that the advantage of GQ will persist. In fact, prior work on autoregressive generation has shown a strong correlation between the generation performance of smaller autoregressive models and that of larger ones, and such smaller models are often used to select tokenizers. For instance, in [GigaTok: Scaling Visual Tokenizers to 3 Billion Parameters for Autoregressive Image Generation], the authors use a small autoregressive model trained with 10× fewer resources to predict the generation performance of a large autoregressive model trained with 64 GPUs. Therefore, we believe our advantage will hold for larger-scale models as well.
>
> ### Q2. How robust is TDC when the target bitrate or codebook size is misspecified relative to the intrinsic per-dimension rates of the Gaussian VAE ?
> * Empirically, TDC is quite robust to mismatches between the codebook size and the bitrate of the Gaussian VAE. As shown in Table 7, when the codebook size is $2^{14}$, the difference in reconstruction quality between Gaussian VAE bitrates of $2^{14}$ and $2^{18}$ is moderate (25.31 vs. 25.24 dB). Nevertheless, we note that it is still advisable to match the codebook size to the bitrate of the Gaussian VAE in accordance with our theoretical results.
>
> ### Q3. Have the authors considered integrating GQ into multi-scale or residual tokenization frameworks ?
> * GQ is a drop-in replacement for VQ and can be integrated into multi-scale or residual tokenization frameworks. We consider the two-stage hierarchical VQ architecture following [Generating Diverse High-Fidelity Images with VQ-VAE-2]. The GQ version simply replaces the VQ layers with Gaussian VAE layers and constructs a two-stage hierarchical Gaussian VAE. It is shown that the TDC achieves accurate bitrate control, and GQ achieves a small quantization error.
>
>     | | bpp | R (stage 1) mean, min-max | R (stage 2) mean, min-max | PSNR | LPIPS | SSIM | rFID |
>     |-|-|-|-|-|-|-|-|
>     | Gaussian VAE w/ TDC (2 stage) | 0.25 + 0.06 ($2^{16}\times 1024 + 2^{16}\times 256$) | 16.11, 14.81-16.98 | 15.69, 15.13-17.12| 28.63 | 0.055 | 0.829 | 0.47 |
>     | GQ (2 stage) | 0.25 + 0.06 ($2^{16}\times 1024 + 2^{16}\times 256$) | - | - | 28.23 | 0.060 | 0.819 | 0.44 |
>
> ### Q4. Could the authors also report precision and recall in Table 6 ?
> * Sure, please find the additional results below. They show that GQ and FSQ remain the best in terms of these metrics.
>
>     | | Codebook Usage | gFID | IS | Precision | Recall |
>     |-|-|------|----|-----------|--------|
>     | Gaussian VAE w/o TDC | - | 8.35 | 202.19 | 0.74 | 0.46 |
>     | Gaussian VAE w/ TDC  | - | 8.47 | 205.94 | 0.77 | 0.49 |
>     | VQGAN | 16.4% | 8.01 | 151.40 | 0.73 | 0.44 |
>     | FSQ | 94.3% | 7.33 | 224.28 | 0.84 | 0.55 |
>     | LFQ | 24.9% | 7.73 | 142.09 | 0.82 | 0.51 |
>     | BSQ | 99.8% | 7.82 | 221.64 | 0.81 | 0.49 |
>     | TokenBridge | 94.6% | 7.82 | 198.24 | 0.80 | 0.51 |
>     | GQ | 100.0% | 7.67 | 230.79 | 0.87 | 0.52 |
>
> We hope these improvements address your concerns. Thank you again for your careful reviews.

---

> > ### Author Rebuttal · Reviewer_aZZv · 2026-04-02
> >
> > I appreciate the authors' efforts in the rebuttal. My concerns have been adequately addressed.

---

> > > ### Author Response · Authors · 2026-04-08
> > >
> > > We greatly thank the reviewer again for their support of our paper. We're glad to be able to address your questions.

---

### Decision · Program_Chairs · 2026-04-30

**Decision:**

Accept (regular)

**Comment:**

In this paper, the authors introduce a new approach to overcome classical issues related to VQ-VAE. They propose to first first train a Gaussian VAE under certain constraints before converting the model into a VQ-VAE without additional training. Reviewers noted that the method is supported by a rigorous mathematical analysis based on the fact that the first-stage Gaussian VAE training relies on a well-understood framework.

During the rebuttal, reviewers raised several concerns. A key issue is that the generative capacity of the model was not fully explored, which raised questions about its effectiveness for generative tasks. The empirical evaluation of generative performance remains somewhat limited. In addition, the paper did not demonstrate performance on very complex tasks. Reviewers also pointed out weaknesses in the presentation: the writing is often dense and difficult to follow, with insufficient explanations on key concepts. Furthermore, the discussion of related work could be expanded.

In response, the authors strengthened their evaluation during the rebuttal by incorporating additional metrics and experiments. They also provided evidence that their approach improves training stability compared to standard methods, while achieving better convergence under the same computational budget. Overall, the paper is technically solid, with a clean methodology and a strong theoretical foundation which I believe is enough even without more complex experiments. Following the discussions, I encourage the authors to carefully address all concerns raised during the rebuttal in the revised version.